# Learning Energy Conserving Dynamics Efficiently with Hamiltonian Gaussian Processes

**Magnus Ross** *magnus.ross@postgrad.manchester.ac.uk*
*University of Manchester*[†]

**Markus Heinonen** *markus.o.heinonen@aalto.fi*
*Aalto University*

Reviewed on OpenReview: *https://openreview.net/forum?id=DHEZuKStzH*

## Abstract

Hamiltonian mechanics is one of the cornerstones of the natural sciences. Recently there has been significant interest in learning Hamiltonian systems in a free-form way directly from trajectory data. Previous methods have tackled the problem of learning from many short, low-noise trajectories, but learning from a small number of long, noisy trajectories, whilst accounting for model uncertainty has not been addressed. In this work, we present a Gaussian process model for Hamiltonian systems with efficient decoupled parameterisation, and introduce an energy-conserving shooting method that allows robust inference from both short and long trajectories. We demonstrate the method's success in learning Hamiltonian systems in various data settings.

## 1 Introduction

Hamiltonian mechanics represent one of the most important classes of dynamical systems, describing wide variety of natural phenomena from electromagnetism to the motion of planets (Salmon, 1988; Taylor, 2005). In this work we consider time-invariant systems characterised by a *Hamiltonian* $\mathcal{H}(\mathbf{q}, \mathbf{p}) \in \mathbb{R}$ over position $\mathbf{q}(t) \in \mathbb{R}^D$ and momenta $\mathbf{p}(t) \in \mathbb{R}^D$ over time $t \in \mathbb{R}_+$, which can be thought of as the total energy of the system's configuration. The rules of evolution of a Hamiltonian system are defined by *Hamilton's equations*

$$\dot{\mathbf{q}} = \frac{\partial \mathcal{H}}{\partial \mathbf{p}}, \qquad \dot{\mathbf{p}} = -\frac{\partial \mathcal{H}}{\partial \mathbf{q}}. \tag{1}$$

We consider problem of learning free-form Hamiltonian $\mathcal{H}(\cdot)$ entirely from observed system trajectories. The conventional mechanistic approach involves manually deriving the Hamiltonian $\mathcal{H}$ and its evolution equations for a system of interest, and possibly estimating system coefficients from data (Seinfeld, 1970; Hernandez & Poznyak, 2020). However, for many systems the Hamiltonian is either unknown or too complex to derive from first principles (Schmidt & Lipson, 2009; Battaglia et al., 2018). Recently, numerous data-driven approaches have been introduced to learn Hamiltonian systems with neural networks (Greydanus et al., 2019; Cranmer et al., 2020; Zhong et al., 2019; Finzi et al., 2020). These methods return point solutions, and are not able to characterise the uncertainty of the solution, which is important when data are limited.

Bayesian approaches, such as Gaussian processes (GPs), can be used to place a prior distribution over the derivative function of a dynamical system, allowing the posterior distribution over the system dynamics to be computed in light of observations (Ridderbusch et al., 2021; Hegde et al., 2022). In this paper we place a GP prior over the Hamiltonian and infer its posterior directly from noisy trajectory data, shown in Figure 1.

---

[†]Work partially completed whilst visiting Aalto University.

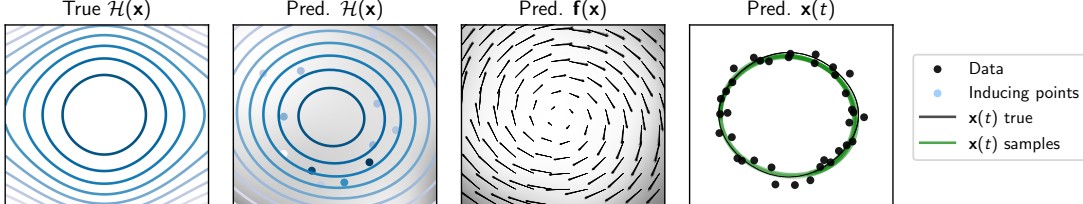

Figure 1: **The proposed model**. We place a GP prior over the Hamiltonian $\mathcal{H}(\mathbf{x})$ and, using a set of inducing points which lie on $\mathcal{H}$, map function samples through Hamilton's equations to obtain system derivative $\mathbf{f}(\mathbf{x})$ samples, to which we apply ODE solver to obtain sample trajectories $\mathbf{x}(t)$. Shading represents model uncertainty.

We develop an energy conserving variational multiple shooting scheme, which allows for efficient inference over long trajectories, which usually present a challenge for dynamical models due to the problem vanishing or exploding gradients (Ribeiro et al., 2020; Metz et al., 2021). Recently, two studies have introduced GP models that also aim to learn the Hamiltonian directly from trajectory data: the symplectic spectrum GP (SSGP) learns the system in a Fourier domain (Tanaka et al., 2022), while the structure preserving GP (SPGP) embeds the model within a numerical symplectic integrator (Ensinger et al., 2022). Both models attempt to skirt the problem of trajectory length by using heuristic methods based on learning from short sub-sequences of the full data, which we find works poorly for systems with complex behaviour.

In this work we present a number of contributions, to both the inference methodology, and the experimental evaluation, of GP models for Hamiltonian systems. Together they can be summarised as follows:

- We propose a Hamiltonian Gaussian process parameterised by inducing variables with efficient functional sampling.

- We adapt the variational multiple shooting method of Hegde et al. (2022) to energy conserving Hamiltonian systems, for highly accelerated optimisation and increased performance.

- We provide an extensive experimental evaluation, and find that our method shows strong performance in a number of settings, whilst discussing the areas in which GP based methods are limited.

## 2    Primer on Hamiltonian mechanics

We consider a Hamiltonian dynamical system over the $2D$-dimensional phase space of canonical positions $\mathbf{q} \in \mathbb{R}^D$ (for example, coordinates or angles) and their associated canonical momenta $\mathbf{p} \in \mathbb{R}^D$. The system is characterised by the Hamiltonian energy function $\mathcal{H}(\mathbf{q}, \mathbf{p}, t) \in \mathbb{R}$ (Thornton & Marion, 2004). In this work we will restrict the discussion to the common case of time-invariant Hamiltonian, i.e. $\mathcal{H}(\mathbf{q}, \mathbf{p}, t) := \mathcal{H}(\mathbf{q}, \mathbf{p})$, in which case, the Hamiltonian energy is conserved over the system trajectories. The temporal evolution of the system is described by a set of coupled first order differential equations, known as Hamilton's equations,

$$\dot{\mathbf{q}} = \frac{d\mathbf{q}}{dt} = \frac{\partial \mathcal{H}}{\partial \mathbf{p}}, \qquad \dot{\mathbf{p}} = \frac{d\mathbf{p}}{dt} = -\frac{\partial \mathcal{H}}{\partial \mathbf{q}}. \tag{2}$$

The structure of Hamilton's equations ensures that various constants of motion are conserved over the system trajectories. Chief among these constants of motion is the energy of the system, but they can also include linear momentum, angular momentum, and other quantities. The energy conservation is shown by

$$\dot{\mathcal{H}} = \frac{d\mathcal{H}}{dt} = \frac{\partial \mathcal{H}}{\partial \mathbf{q}}\dot{\mathbf{q}} + \frac{\partial \mathcal{H}}{\partial \mathbf{p}}\dot{\mathbf{p}} = \frac{\partial \mathcal{H}}{\partial \mathbf{q}}\frac{\partial \mathcal{H}}{\partial \mathbf{p}} - \frac{\partial \mathcal{H}}{\partial \mathbf{p}}\frac{\partial \mathcal{H}}{\partial \mathbf{q}} = 0. \tag{3}$$

By using Hamilton's equations we obtain a system of $2D$ first order differential equations,

$$\frac{d\mathbf{x}}{dt} = \mathbf{f}(\mathbf{x}) = \begin{pmatrix} \frac{\partial \mathcal{H}}{\partial \mathbf{p}} \\ -\frac{\partial \mathcal{H}}{\partial \mathbf{q}} \end{pmatrix}, \qquad \mathbf{x} = \begin{pmatrix} \mathbf{q} \\ \mathbf{p} \end{pmatrix} \in \mathbb{R}^{2D} \tag{4}$$

where we have concatenated $\mathbf{q}$ and $\mathbf{p}$ into the $2D$ dimensional phase space state vector $\mathbf{x}$, and $\mathbf{f} : \mathbb{R}^{2D} \mapsto \mathbb{R}^{2D}$ is the system time differential. We obtain system trajectories by forward integration

$$\mathbf{x}(t) := \mathbf{x}(t; \mathbf{x}_0) = \mathbf{x}_0 + \int_0^t \mathbf{f}(\mathbf{x}(\tau)) d\tau, \tag{5}$$

where $\mathbf{x}(t)$ is the state of the system at time $t$, $\mathbf{x}_0$ is the initial state of the system, and $\tau \in [0, t]$ is an integration time variable. Typically for mechanical systems the Hamiltonian is determined by identifying the kinetic and potential energies of the constituent parts, which form the total energy $\mathcal{H}$, with the process becoming increasingly difficult for more complex systems. In the this work, we aim to forgo this process and to learn the Hamiltonian directly from trajectory data, with no assumptions on its functional form.

## 3 Hamiltonian Gaussian processes

In order to infer the Hamiltonian of the system, we assume it follows a Gaussian process (GP) prior (For review, see Williams & Rasmussen (2006))

$$\mathcal{H}(\mathbf{x}) \sim \mathcal{GP}(0, k_{\mathcal{H}}(\mathbf{x}, \mathbf{x}')), \tag{6}$$

which models a distribution over energy surfaces with zero mean $\mathbb{E}[\mathcal{H}(\mathbf{x})] = 0$ and kernelised covariance,

$$\text{cov}[\mathcal{H}(\mathbf{x}), \mathcal{H}(\mathbf{x}')] = k_{\mathcal{H}}(\mathbf{x}, \mathbf{x}'), \qquad k_{\mathcal{H}} : \mathbb{R}^{2D} \times \mathbb{R}^{2D} \mapsto \mathbb{R}. \tag{7}$$

In a GP the probability of a function at any finite subset of evaluations follows a multivariate Gaussian

$$(\mathcal{H}(\mathbf{x}_1), \ldots, \mathcal{H}(\mathbf{x}_N)) \sim \mathcal{N}(\mathbf{0}, k_{\mathcal{H}}(\mathbf{X}, \mathbf{X})), \tag{8}$$

where $k_{\mathcal{H}}(\mathbf{X}, \mathbf{X}) \in \mathbb{R}^{N \times N}$ is a positive definite kernel matrix with elements $[k_{\mathcal{H}}(\mathbf{X}, \mathbf{X})]_{ij} = k_{\mathcal{H}}(\mathbf{x}_i, \mathbf{x}_j)$. In order to perform inference whilst using a GP representation of the Hamiltonian, we adapt the method described by Hegde et al. (2022), which allows for inference of ODE systems with a GP based derivative function. This is possible because a GP prior over the Hamiltonian implies a GP prior over the derivative function, as we will see in the following section.

### 3.1 The time derivative

Given the prior over $\mathcal{H}$ it is necessary to define the system time derivative $\dot{\mathbf{x}}$, in order to compute trajectories for inference and sampling. Equation (4) can be rewritten (Rath et al., 2021) as

$$\dot{\mathbf{x}} = \mathbf{f}(\mathbf{x}) = \mathcal{L}\mathcal{H}(\mathbf{x}), \quad \text{with} \quad \mathcal{L} = \underbrace{\begin{pmatrix} 0 & I \\ -I & 0 \end{pmatrix}}_{\text{Poisson tensor}} \nabla_{\mathbf{x}} = \begin{pmatrix} \partial_p \\ -\partial_q \end{pmatrix}. \tag{9}$$

Gaussian processes are closed under linear operators (Williams & Rasmussen, 2006; Agrell, 2019), and hence the Hamiltonian and its vector field follow a zero-mean joint GP

$$\begin{pmatrix} \mathcal{H}(\mathbf{x}) \\ \mathbf{f}(\mathbf{x}) \end{pmatrix} \sim \mathcal{GP}\left(0, \begin{pmatrix} k_{\mathcal{H}}(\mathbf{x}, \mathbf{x}') & \mathbf{k}_{\mathcal{H}\mathbf{f}}(\mathbf{x}, \mathbf{x}') \\ \mathbf{k}_{\mathbf{f}\mathcal{H}}(\mathbf{x}, \mathbf{x}') & K_{\mathbf{f}}(\mathbf{x}, \mathbf{x}') \end{pmatrix}\right), \tag{10}$$

with covariances induced by the Hamilton's equation,

$$\text{cov}\left[\mathcal{H}(\mathbf{x}), \mathbf{f}(\mathbf{x}')\right] = \mathbf{k}_{\mathcal{H}\mathbf{f}}(\mathbf{x}, \mathbf{x}') = \begin{pmatrix} \partial_p \\ -\partial_q \end{pmatrix} k_{\mathcal{H}}(\mathbf{x}, \mathbf{x}') \qquad \in \mathbb{R}^{2D \times 1} \tag{11}$$

$$\text{cov}\left[\mathbf{f}(\mathbf{x}), \mathbf{f}(\mathbf{x}')\right] = K_{\mathbf{f}}(\mathbf{x}, \mathbf{x}') = \begin{pmatrix} \partial_{pp}^2 & -\partial_{pq}^2 \\ -\partial_{qp}^2 & \partial_{qq}^2 \end{pmatrix} k_{\mathcal{H}}(\mathbf{x}, \mathbf{x}') \qquad \in \mathbb{R}^{2D \times 2D}. \tag{12}$$

## 3.2 Inducing points

In order to allow tractable inference and sampling, we introduce a set of inducing points (Snelson & Ghahramani, 2006) to obtain a finite, parametric representation of the infinite joint GP $(\mathcal{H}, \mathbf{f})$. We condition the Hamiltonian with $M$ inducing energies $\mathbf{u} = (u_1, \ldots, u_M) \in \mathbb{R}^M$ at phase space locations $\mathbf{Z} = (\mathbf{z}_1, \ldots, \mathbf{z}_M) \in \mathbb{R}^{M \times 2D}$ that encode energy pseudo-observations $u = \mathcal{H}(\mathbf{z})$. The vector field conditioned on these pseudo-observations is again a Gaussian process,

$$\mathbf{f}(\mathbf{x}) \big| \big(\mathcal{H}(\mathbf{Z}) = \mathbf{u}\big) \sim \mathcal{GP}\Big( \underbrace{\mathbf{k}_{\mathcal{H}\mathbf{f}}(\mathbf{x}, \mathbf{Z}) k_{\mathcal{H}}(\mathbf{Z}, \mathbf{Z})^{-1} \mathbf{u}}_{\text{mean}}, \underbrace{K_{\mathbf{f}}(\mathbf{x}, \mathbf{x}') - \mathbf{k}_{\mathbf{f}\mathcal{H}}(\mathbf{x}, \mathbf{Z}) k_{\mathcal{H}}(\mathbf{Z}, \mathbf{Z})^{-1} \mathbf{k}_{\mathcal{H}\mathbf{f}}(\mathbf{Z}, \mathbf{x}')}_{\text{covariance}} \Big), \tag{13}$$

where the set inputs refer to expanding the corresponding function over them. By varying the inducing parameters $(\mathbf{u}, \mathbf{Z})$, we can represent approximately arbitrary Hamiltonians and their unique vector fields, which can then be forward integrated to obtain simulated trajectory solutions. In later sections these parameters are the main variables to be learnt. Our approach differs from Hegde et al. (2022) by placing the inducing points on the Hamiltonian, not the derivative function. We then require a set of scalar inducing points, instead of vector-valued $2D$-dimensional inducing points.

## 3.3 Sampling

In order to simulate trajectories $\mathbf{x}(0), \ldots, \mathbf{x}(T)$ we must be able to sample a persistent derivative function or vector field $\dot{\mathbf{x}} = \mathbf{f} \sim \mathcal{GP}$ from the GP (13), and evaluate it along the trajectory $\mathbf{x}(t)$. Standard methods for sampling functions from a GP are based on kernel matrix decompositions, which have prohibitive complexity of $\mathcal{O}(N^3)$ for $N$ evaluation points. We follow Hegde et al. (2022), and bypass this problem by a 'decoupled' parameterisation (Wilson et al., 2020), sampling from the Hamiltonian instead of the derivative function,

$$\mathcal{H}_{\tilde{\mathbf{w}}, \mathbf{u}, \mathbf{Z}}(\mathbf{x}) = \sum_{i=1}^{S} w_i \phi_i(\mathbf{x}) + \sum_{j=1}^{M} \nu_j k(\mathbf{x}, \mathbf{z}_j), \tag{14}$$

where $w_i \sim \mathcal{N}(0, 1)$ are random weights of the $S$ Fourier basis functions $\phi_i(\mathbf{x}) = \cos(\boldsymbol{\alpha}_i^\top \mathbf{x} + \beta_i)$ with frequencies $\boldsymbol{\alpha}_i$ sampled from the spectral density of $k$, and $\beta_i \sim U(0, 2\pi)$ (Rahimi & Recht, 2007), and $\boldsymbol{\nu} = k(\mathbf{Z}, \mathbf{Z})^{-1}(\mathbf{u} - \boldsymbol{\Phi}\mathbf{w})$ where $\boldsymbol{\Phi} = \phi(\mathbf{Z}) \in \mathbb{R}^{M \times S}$ represents the evaluation of bases over inducing inputs. We denote all Fourier-related parameters by $\tilde{\mathbf{w}} = \{\mathbf{w}, \boldsymbol{\alpha}, \boldsymbol{\beta}\}$. By sampling and fixing $\tilde{\mathbf{w}}$, we obtain a deterministic function sample that can be evaluated anywhere. Finally, we transform the energy surface $\mathcal{H}$ to a derivative vector field via $\mathbf{f}_{\tilde{\mathbf{w}}, \mathbf{u}, \mathbf{Z}}(\mathbf{x}) = \mathcal{L}\mathcal{H}_{\tilde{\mathbf{w}}, \mathbf{u}, \mathbf{Z}}(\mathbf{x})$, which allows trajectory rollout using numerical integration.

## 3.4 The probabilistic model

We aim to infer $\mathcal{H}$ from one or more noisy realisations of trajectories from the true system, with arbitrary length and time irregularity, each of which have different unknown initial conditions. We present simplified notation for a single observed trajectory (See Appendix A.3 for multiple trajectory derivations).

Let the trajectory observation be denoted $\mathbf{Y} = (\mathbf{y}_1, \ldots \mathbf{y}_N)^T \in \mathbb{R}^{N \times 2D}$, where $\mathbf{y}_i = \mathbf{x}_{\text{true}}(t_i) + \boldsymbol{\epsilon}_i \in \mathbb{R}^{2D}$ is the $i$'th noisy state of the system at time $t_i \in (t_1, \ldots, t_N)$, and $N$ is the number of time steps observed. We begin by assuming a Gaussian prior on the inducing energies and on the initial state $\mathbf{x}_0$,

$$p(\mathbf{u}) = \mathcal{N}(\mathbf{u}|\mathbf{0}, k(\mathbf{Z}, \mathbf{Z})) \tag{15}$$

$$p(\mathbf{x}_0) = \mathcal{N}(\mathbf{x}_0|\mathbf{0}, \mathbf{I}), \tag{16}$$

where $k(\mathbf{Z}, \mathbf{Z}) \in \mathbb{R}^{M \times M}$ is covariance matrix for the inducing states, with $[k(\mathbf{Z}, \mathbf{Z})]_{i,j} = k_{\mathcal{H}}(\mathbf{z}_i, \mathbf{z}_j)$. We follow the convention in sparse Gaussian processes to treat the inducing locations $\mathbf{Z}$ as hyperparameters that are merely optimised, instead of inferring their posterior distribution (Hensman et al., 2015a) (For in-depth discussion, see Hensman et al. (2015b); Rossi et al. (2021)). We omit all hyperparameters from the probabilistic model notation below for simplicity.

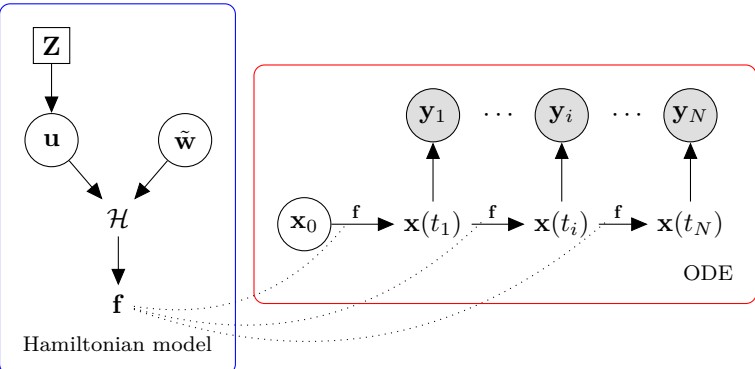

Figure 2: Plate diagram over the model (blue) and ODE system (red). We denote observations $\{\mathbf{y}_i\}$ with shaded nodes, random variables $(\tilde{\mathbf{w}}, \mathbf{u}, \mathbf{x}_0)$ as white nodes, hyperparameters $(\mathbf{Z})$ with rectangles, and functions $(\mathcal{H}, \mathbf{f}, \mathbf{x})$ without shapes. Both sides contain the same $\mathbf{f}$.

The joint distribution over data $\mathbf{Y}$, sampling parameters $\tilde{\mathbf{w}}$, energy variables $\mathbf{u}$ and initial state $\mathbf{x}_0$ is

$$p(\mathbf{Y}, \tilde{\mathbf{w}}, \mathbf{u}, \mathbf{x}_0) = p(\mathbf{Y}|\tilde{\mathbf{w}}, \mathbf{u}, \mathbf{x}_0)p(\tilde{\mathbf{w}})p(\mathbf{u})p(\mathbf{x}_0) \tag{17}$$

$$= \prod_{i=1}^{N}\Big[p(\mathbf{y}_i|\tilde{\mathbf{w}}, \mathbf{u}, \mathbf{x}_0)\Big]p(\tilde{\mathbf{w}})p(\mathbf{u})p(\mathbf{x}_0), \tag{18}$$

where the likelihood is

$$p(\mathbf{y}_i|\tilde{\mathbf{w}}, \mathbf{u}, \mathbf{x}_0) = \mathcal{N}\Big(\mathbf{y}_i\big|\mathbf{x}(t), \sigma_{\text{obs}}^2 I\Big), \qquad \mathbf{x}(t) := \mathbf{x}_{\tilde{\mathbf{w}}, \mathbf{u}, \mathbf{x}_0}(t) \tag{19}$$

where we first compute the Hamiltonian $\mathcal{H}$ with Equation (14), then its derivative $\mathbf{f}$ using Equation (4), and finally integrate forward $\mathbf{x}_0 \xrightarrow{\mathbf{f}} \mathbf{x}(t)$ with Equation (5). The form of the joint distribution here mirrors Hegde et al. (2022). Our goal is to infer the intractable posterior

$$p(\mathbf{u}, \mathbf{x}_0|\mathbf{Y}) = \int p(\mathbf{u}, \mathbf{x}_0|\mathbf{Y}, \tilde{\mathbf{w}})p(\tilde{\mathbf{w}})d\tilde{\mathbf{w}}, \tag{20}$$

over initial state $\mathbf{x}_0$ and inducing variables $\mathbf{u}$, where we assume the tractable sampling variables to be $\tilde{\mathbf{w}}$ marginalised. Figure 2 shows the dependency structure between the variables in the model in graphical form. We turn to variational inference to approximate the posterior.

### 3.5 Variational inference

We follow Hegde et al. (2022) and use the framework of stochastic VI (Hoffman et al., 2013; Hensman et al., 2013) to infer the posterior of Equation (20). We assume a factorised posterior approximation

$$q(\mathbf{u}, \mathbf{x}_0) = q(\mathbf{u})q(\mathbf{x}_0) \tag{21}$$

$$q(\mathbf{u}) = \mathcal{N}(\mathbf{u}|\mathbf{m}, \mathbf{Q}) \tag{22}$$

$$q(\mathbf{x}_0) = \mathcal{N}(\mathbf{x}_0|\mathbf{m}_0, \mathbf{Q}_0), \tag{23}$$

with variational free parameters $\theta = (\mathbf{m}, \mathbf{Q}, \mathbf{m}_0, \mathbf{Q}_0)$. Variational inference seeks to find $\arg\min_\theta \text{KL}\big[q_\theta(\mathbf{u}, \mathbf{x}_0)\,\|\,p(\mathbf{u}, \mathbf{x}_0|\mathbf{Y})\big]$, the Kullback-Leibler divergence between the approximate posterior $q$ and the true posterior, which is equivalent to maximising the evidence lower bound (Blei et al., 2017)

$$\mathcal{F}(\theta, \mathbf{Z}) = \mathbb{E}_{q(\mathbf{u})q(\mathbf{x}_0)p(\tilde{\mathbf{w}})}\left[\sum_{i=1}^{N}\log p(\mathbf{y}_i|\tilde{\mathbf{w}}, \mathbf{u}, \mathbf{x}_0)\right] - \text{KL}\big[q(\mathbf{u})\|p(\mathbf{u})\big] - \text{KL}\big[q(\mathbf{x}_0)\|p(\mathbf{x}_0)\big]. \tag{24}$$

The expectations integrate the inducing energies $\mathbf{u}$, initial state $\mathbf{x}_0$ estimates, and Fourier function sampling determined by $\tilde{\mathbf{w}}$. The likelihood requires solving the trajectory $\mathbf{x}_{\tilde{\mathbf{w}},\mathbf{u},\mathbf{x}_0}(t)$ numerically. The expectations are Monte Carlo averaged, while the KL terms have closed-form solutions. The bound is maximised by gradient ascent wrt $\theta$ and $\mathbf{Z}$. See appendix A.1 for details.

## 4 Energy-conserving shooting parallelisation

Optimisation of the variational bound (24) for long sequences is challenging in practice due to the problem of unstable gradients (Ribeiro et al., 2020; Metz et al., 2021). For long trajectories, small perturbations at early times can compound into large effects at later times, resulting in gradients of the bound vanishing or exploding (Haber & Ruthotto, 2017). Although this problem has primarily been discussed in the context of neural networks (Kim et al., 2021; Choromanski et al., 2020), Hegde et al. (2022) introduces a probabilistic shooting solution for GP-ODEs. Shooting is an ODE optimisation technique where the 'long' solution $\mathbf{x}(0) \mapsto \mathbf{x}(T)$ is split into consecutive segments $[\mathbf{x}(t_l), \mathbf{x}(t_{l+1}))$ that are solved in parallel, while ensuring that the neighboring segments match (Hemker, 1974; Bock & Plitt, 1984) (For review, see Diehl & Gros (2020)).

### 4.1 Energy-conserving shooting model

We adapt the shooting formulation of Hegde et al. (2022) to Hamiltonian GP ODEs, in order to stabilise gradients and allow learning over longer sequences. We begin by augmenting the system with a set of $L < N$ shooting variables $\mathbf{S} = \{\mathbf{s}_l\}_{l=0}^L \in \mathbb{R}^{L \times 2D}$, which represent the state of the system at times $t_l \in \{t_l\}_{l=0}^L$ and split the continuous state solution $\mathbf{x}(t; \mathbf{x}_0)$ into $L$ distinct segments from initial states $\mathbf{s}_l$ with segment solutions

$$\mathbf{x}(t; \mathbf{s}_l) = \mathbf{s}_l + \int_{t_l}^{t_{l+1}} \mathbf{f}(\mathbf{x}(\tau))d\tau, \qquad \text{for } t \in [t_l, t_{l+1}]. \tag{25}$$

That is, to solve a particular time $t$, we need to find its interval $[t_l, t_{l+1}]$ and solve from $\mathbf{s}_l$. The splitting leads to more stable gradients since each segment has less non-linear solution map (Diehl & Gros, 2020). Furthermore, the segments can be solved in parallel. The key problem of shooting is to ensure continuity $\mathbf{x}(t_l; s_{l-1}) = \mathbf{s}_l$ at segment boundaries $t_l$, otherwise the combined solution will be discontinuous.

We formulate this with an error model

$$\mathbf{s}_l = \mathbf{x}(t_l; \mathbf{s}_{l-1}) + \boldsymbol{\xi}_l, \qquad \boldsymbol{\xi}_l \sim \mathcal{N}(\mathbf{0}, \sigma_\xi^2 I) \tag{26}$$

where $\boldsymbol{\xi}_l \in \mathbb{R}^{2D}$ is the between-segment tolerance of position and momenta. To conserve energy, we also introduce an energy tolerances $\chi$ to encode the permissible energy change between segments,

$$\mathcal{H}(\mathbf{s}_l) = \mathcal{H}\big(\mathbf{x}(t_l; \mathbf{s}_{l-1})\big) + \chi_l, \qquad \chi_l \sim \mathcal{N}(0, \sigma_\chi^2). \tag{27}$$

Figure 3: The shooting system.

Together these constraints lead us to a product prior

$$p(\mathbf{s}_l|\mathbf{s}_{l-1}, \tilde{\mathbf{w}}, \mathbf{u}) = \mathcal{N}\big(\mathbf{s}_l|\mathbf{x}(t_i; \mathbf{s}_{l-1}), \sigma_\xi^2 \mathbf{I}\big) \mathcal{N}\Big(\mathcal{H}(\mathbf{s}_l)|\mathcal{H}\big(\mathbf{x}(t_i; \mathbf{s}_{l-1})\big), \sigma_\chi^2\Big), \qquad \mathcal{H}(\cdot) := \mathcal{H}_{\tilde{\mathbf{w}},\mathbf{u}}(\cdot) \tag{28}$$

where $\mathcal{H}$ is the Hamiltonian function conditioned by the $\tilde{\mathbf{w}}, \mathbf{u}$, and $\mathbf{x}(t_l; \mathbf{s}_{l-1})$ depends on $\mathcal{LH}$ via Equations (4) and (5). We place a Gaussian prior on the initial state, $p(\mathbf{s}_0) = \mathcal{N}(\mathbf{s}_0|\mathbf{0}, \mathbf{I})$. Let $\mathbf{s}_{l(i)}$ denote the last shooting variable before time $t_i$. The joint distribution of the shooting-augmented model is

$$p(\mathbf{Y}, \tilde{\mathbf{w}}, \mathbf{u}, \mathbf{S}) = \underbrace{\left[\prod_{i=1}^N p(\mathbf{y}_i|\tilde{\mathbf{w}}, \mathbf{u}, \mathbf{s}_{l(i)})\right]}_{\text{shooting likelihood}} \underbrace{\left[\prod_{l=1}^L p(\mathbf{s}_l|\mathbf{s}_{l-1}, \tilde{\mathbf{w}}, \mathbf{u})\right]}_{\text{tolerance prior}} p(\tilde{\mathbf{w}})p(\mathbf{u})p(\mathbf{s}_0). \tag{29}$$

## 4.2 Variational shooting inference

The inference of the posterior $p(\mathbf{u}, \mathbf{S}|\mathbf{Y})$ is again intractable. We define a variational posterior approximation $q(\mathbf{u}, \mathbf{S}) = q(\mathbf{u}) \prod_{l=0}^{L} q(\mathbf{s}_l)$ with independent Gaussians $q(\mathbf{s}_l) = \mathcal{N}(\mathbf{s}_l|\mathbf{a}_l, \Sigma_l)$ on the shooting states. This results in new evidence lower bound

$$\mathcal{F}(\theta, \mathbf{Z}) = \mathbb{E}_{q(\mathbf{u})p(\tilde{\mathbf{w}})} \left[ \sum_{i=1}^{N} \mathbb{E}_{q(\mathbf{s}_{l(i)})} \left[ \log p(\mathbf{y}_i|\tilde{\mathbf{w}}, \mathbf{u}, \mathbf{s}_{l(i)}) \right] + \sum_{l=1}^{L} \mathbb{E}_{q(\mathbf{s}_l)q(\mathbf{s}_{l-1})} \left[ \log p(\mathbf{s}_l|\mathbf{s}_{l-1}, \tilde{\mathbf{w}}, \mathbf{u}) \right] \right] \tag{30}$$
$$- \sum_{l=1}^{L} \mathbb{H}[q(\mathbf{s}_l)] - \mathrm{KL}[q(\mathbf{s}_0) \,||\, p(\mathbf{s}_0)] - \mathrm{KL}[q(\mathbf{u}) \,||\, p(\mathbf{u})],$$

where $\theta = (\mathbf{m}, \mathbf{Q}, \{\mathbf{a}_l, \Sigma_l\})$. The first expectation is the 'short' likelihood of each observation solved only from previous shooting state. The second term ensures the quality of the match between the shooting segments. The remaining terms regularise the model. This bound is similar to that of Hegde et al. (2022), but places the inducing points on the Hamiltonian, not the derivative function, and includes an additional energy based matching term for the shooting states. See appendix A.2 for details.

# 5   Related work

Methodologically our work is founded on Hegde et al. (2022), who present learning of non-Hamiltonian ODEs with GPs using variational multiple shooting for efficient long trajectory inference. We extend with Hamiltonian structure on both the system and the shooting method to control energy conservation.

**Hamiltonian Gaussian processes.**   The most similar existing work to ours are the symplectic spectrum GP (SSGP) model of Tanaka et al. (2022), published concurrently, and the structure-preserving GP (SPGP) model of Ensinger et al. (2022), both of which place a GP prior over the Hamiltonian function $\mathcal{H}$. The SSGP model is closely related to ours, but differs in inference. They use standard RFFs for sampling, which suffers from variance starvation and poor uncertainty representation (Wilson et al., 2020), and place a variational distribution on the RFF weights instead of inducing energies. The SPGP proposes to embed a Hamiltonian GP specifically within a symplectic integrator to ensure numerical volume preservation. Neither model supports initial state estimation and neither use shooting approximations, but instead resort to heuristic minibatching of subsequences of the trajectories.

**Hamiltonian maps with Gaussian processes.**   A number of works aim to learn the Hamiltonian flow map $\mathbf{x}(t_{\mathrm{init}}) \mapsto \mathbf{x}(t_{\mathrm{final}})$, or related mappings, from initial and final conditions. These methods typically require large number of low-noise trajectories (10s or 100s). Rath et al. (2021) use GPs to learn the flow map of Hamiltonian systems, providing an implicit learning scheme for non-separable systems, and an more efficient explicit scheme for separable systems. Offen & Ober-Blöbaum (2022) develop a scheme known as shadow symplectic integration (SSI), which allows for compensation for the error incurred by forward numerical integration, leading to more accurate preservation of symplectic structure. Bertalan et al. (2019) learn the Hamiltonian directly in phase space, which requires an approximation of the time derivatives of the coordinates using first differences, introducing additional error, especially for noisy data.

**Hamiltonian neural networks.**   There has been a significant amount of interest in learning Hamiltonian dynamics with neural networks (NNs). Greydanus et al. (2019) introduce the Hamiltonian NN (HNN), which trains an NN to predict the dynamics with an auxiliary loss based on Hamilton's equations. The original formulation of the HNN requires computation of the time derivatives of the coordinates, and cannot learn directly from trajectory data. Zhong et al. (2019) introduce a version of the HNN which computes trajectory rollouts using an ODE solver and can learn directly from trajectories, in addition to incorporating control signals. Finzi et al. (2020) introduce constrained HNNs for mechanical systems, in which the structure of the system is encoded as constraints, and the Hamiltonian is learned in Cartesian coordinates. Gruver et al. (2021) investigate the effect of the inductive biases in Hamiltonian and Lagrangian NNs, and find that for realistic systems, baseline models without energy conservation often perform better.

## 6  Experiments

In this section we provide an experimental evaluation of our method for a variety of Hamiltonian systems. The code for our implementation of the model is available at `https://github.com/magnusross/hgp`.

### 6.1  Experimental Setup

**Tasks.**  Aside from an initial illustrative example, we study two distinct tasks:

- **Task 1: Trajectory forecasting.** In this task, the model is provided with a single noisy trajectory of length $T$ with unknown initial condition. The model learns the system from $[0, T]$ interval, and is tasked to forecast the trajectory forward for $[T, 2T]$.

- **Task 2: Initial condition extrapolation.** In this task, the model is provided with $K$ noisy trajectories of fixed length from different, unknown initial conditions. The model is evaluated on its ability to forecast trajectories from a new set of noise-free initial conditions.

Most prior work learning Hamiltonians deal with variations of task 2, for example Tanaka et al. (2022); Rath et al. (2021). Experiments for task 1 are provided by Ensinger et al. (2022) and Hegde et al. (2022) among others, but no prior work has provided a joint evaluation of both tasks. For both tasks we report the trajectory and Hamiltonian energy root mean square error (RMSE), and the mean negative log likelihood (MNLL). We repeat each experiment 10 times with different initial conditions, and report median and interquartile range on all tables and plots due to the high variability of the results for all models. We give tables of means and standard errors in appendix C.

**Systems.**  We evaluate our model on three true Hamiltonian systems: the fixed pendulum (FP), the spring pendulum (SP) (Lynch, 2000), and the Henon-Heiles (HH) system (Henon & Heiles, 1964). The FP is two-dimensional with predictable periodic dynamics, and serves as a simple test case. The SP and HH are both four-dimensional, and exhibit more complex, chaotic dynamics. The system definitions are

$$\text{(fixed pendulum)} \qquad \mathcal{H}_{\text{FP}}(q, p) = mgr(1 - \cos q) + \frac{1}{2mr^2}p^2$$

$$\text{(spring pendulum)} \qquad \mathcal{H}_{\text{SP}}(q_1, q_2, p_1, p_2) = \frac{1}{2m}\left(p_1^2 + \frac{p_2^2}{(q_1 + r)^2}\right) + \frac{1}{2}kq_1^2 - mgr\cos q_2$$

$$\text{(Henon-Heiles)} \qquad \mathcal{H}_{\text{HH}}(q_1, q_2, p_1, p_2) = \frac{1}{2}\left(q_1^2 + q_2^2 + p_1^2 + p_2^2\right) + \mu\left(q_2 q_1^2 - \frac{1}{3}q_2^3\right),$$

where $m$ is the pendulum mass, $r$ is its resting length, $g$ is the gravitational field strength, $k$ is the spring constant, and $\mu$ is a parameter controlling the magnitude of the HH potential. For more details on each system, see the appendix B.1. We add Gaussian noise to the training data for each task, with variance set to 5% of signal variance, unless otherwise mentioned. For each experiment we sample different initial conditions from the phase space of the system to generate data for each repeat, using the same data across each model we test.

**Our HGP setup.**  We use $M = 48$ inducing points for task 1, and $M = 128$ for task 2. Throughout we use $S = 256$ basis functions, and run optimisation for 2500 iterations using the Adam optimiser (Kingma & Ba, 2015) with learning rate $3e^{-3}$. During training we use a single sample from the model to estimate the intractable expectations, when making predictions we use 32 samples. Unless otherwise stated, we use the shooting approximation with a single shooting state per 4 data points, i.e. $L = \lfloor N/4 \rfloor$. We use the `torchdiffeq` package (Chen, 2018) with implicit `dopri5` solver. For more details on the setup of the HGP and all the baseline models see appendix B.3.

**Hamiltonian aware initialisation.**  We optimise the bounds in Equations (24) and (30) using gradient ascent on the variational parameters, and the set of model hyperparameters. This optimisation problem is

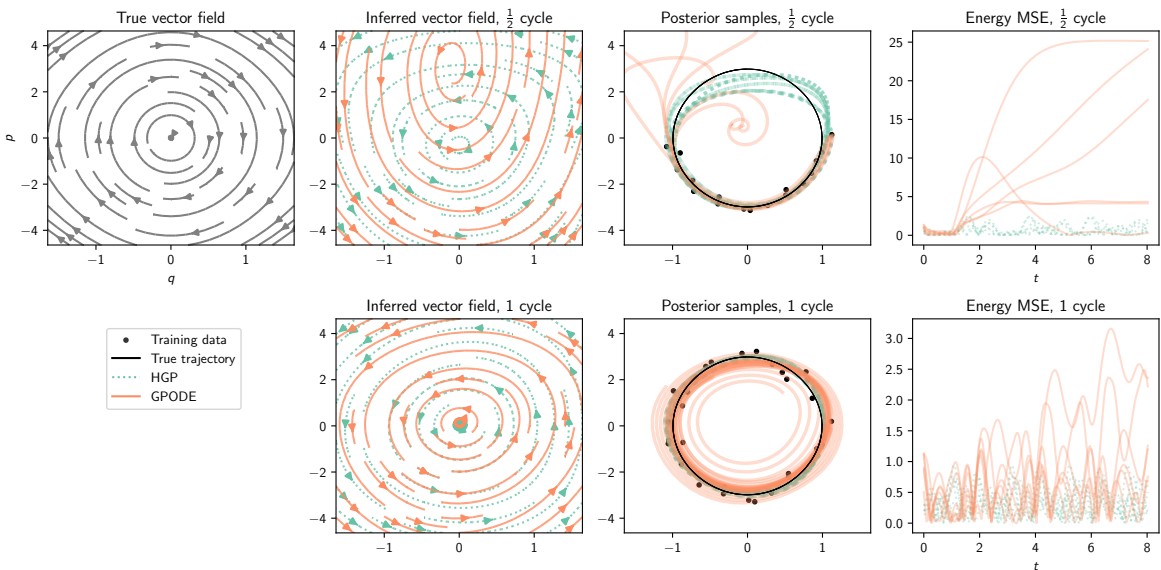

Figure 4: **The HGP can accurately model energy-preserving Hamiltonians**. Top row: the true pendulum system (FP) followed by learnt vector field of HGP (orange) and non-Hamiltonian GPODE baseline (green) from half a cycle of data, followed by sample trajectories and the error in predicted energy along the trajectory. Bottom row: full cycle estimates, where the baseline is able to fit the vector field, but is oblivious to the energy.

challenging, as such good initialisations are very important, particularly for the variational parameters of the Hamiltonian GP. We can obtain a good initialisation by observing that the derivative function $\mathbf{f}$ and $\mathcal{H}$ are jointly a GP. Using the approximate numerical derivatives of the trajectory data, we can form some data for $\mathbf{f}$, which can then be conditioned on, using equations (10) to (12), to obtain an estimate for the mean function of the Hamiltonian. We use this estimate as the initial value for the variational mean $\mathbf{m}$. See appendix B.2 for details.

**NN baselines.** We compare against two NN models: a standard (non-Hamiltonian) Neural ODE (NODE) (Chen et al., 2018) with a neural time derivative $\mathbf{f}_{\boldsymbol{\theta}}(\mathbf{x})$, and a HNN, based on the Unstructured SymODEN model of Zhong et al. (2019) with a neural Hamiltonian $\mathcal{H}_{\boldsymbol{\theta}}(\mathbf{x})$. Both methods perform poorly when trained on raw trajectories due to vanishing/exploding gradients. To allow for a fair comparison, we split the trajectories into smaller sub-trajectories by running a sliding window of fixed length over the data, and form these into batches of training data.

**GP baselines.** In order to evaluate the effect of our Hamiltonian prior, we compare against the non-Hamiltonian shooting GPODE baseline of Hegde et al. (2022). We use the same hyperparameter settings and initialisations as in the HGP model, where applicable. To initialise the inducing variable distribution means, we follow the same procedure described by Hegde et al. (2022). We are unable to provide results for the SPGP (Ensinger et al., 2022) as the authors did not release their code publicly, and would not make an implementation available on request. A public code exists for the SSGP, however it does not support systems with $D > 1$. In order to provide some comparison we re-implement one of the key points of difference between the models, the sub-sequence training method, and test its performance. We also provide a comparison with the SSGP for the toy FP system in Appendix C.3.

## 6.2 Toy task

As an initial test of the effect of the Hamiltonian prior we train both HGP and GPODE, without shooting, on the FP system with both half a cycle and a full cycle of data. Figure 4 shows the inferred vector fields,

the posterior trajectory samples, and the error in the energy of these trajectories. We see the HGP producing plausible trajectories and appropriate vector field estimate even far from the observations when given half a cycle of training data, whilst the GPODE does not, and produces samples with a large energy violation. With full cycle of data both models fit the data well, however the GPODE still has high energy violations. This toy task illustrates the ability of the HGP to learn accurate dynamics from a relatively small amount of data by its Hamiltonian inductive bias.

### 6.3 Task 1: trajectory forecasting

| Method | State RMSE ($\downarrow$) | | | State MNLL ($\downarrow$) | | | Energy RMSE ($\downarrow$) | | |
|---|---|---|---|---|---|---|---|---|---|
| | FP | HH | SP | FP | HH | SP | FP | HH | SP |
| NODE | **0.12** (0.12) | 0.42 (0.25) | 1.13 (0.82) | - | - | - | **0.30** (0.28) | 0.02 (0.01) | 1.02 (0.41) |
| HNN | 0.20 (0.16) | 1.56 (0.26) | 1.41 (0.30) | - | - | - | 0.34 (0.33) | 0.04 (0.10) | 1.12 (0.25) |
| GPODE | 0.23 (0.26) | 0.54 (0.71) | 0.91 (0.36) | -0.03 (1.21) | 1.20 (4.19) | 2.77 (2.77) | 0.38 (0.34) | 0.02 (0.02) | **0.78** (0.50) |
| HGP | 0.18 (0.19) | **0.32** (0.30) | **0.64** (0.50) | **-0.21** (0.67) | **0.31** (1.89) | **1.39** (2.20) | 0.25 (0.42) | **0.01** (0.01) | 0.81 (0.36) |

Table 1: Performance comparison of different methods on each system on the trajectory forecasting task.

Table 1 compares the performance of HGP to baseline methods for each of system on the forecasting task. We use train trajectory lengths of 40, 8 and 16 seconds for the HH, FP and SP systems respectively.[1] For the more complex systems (SP, HH) the HGP provides strong performance particularly when compared to the equivalent GP baseline, the GPODE, providing significantly better point predictions and uncertainty quantification. For the simple FP system the NODE provides a similar result to the HGP, for the SP and HH systems the HGP provides the best performance. In this and subsequent experiments, we found that the NODE unexpectedly outperformed the HNN, this issue is discussed in Section 7. Table 1 additionally shows the energy RMSE, computed by applying the true Hamiltonian to the predicted trajectories, showing that the HGP is able to recover the energy conserving dynamics well. Figure 5 shows the cumulative errors accrued over the test period. Appendix C contains plots of the model predictions for a sample trajectory from each system.[2] These animations are illuminating because they show that, in addition to providing strong predictive performance, the GP models produce trajectories that are more physically plausible than the NN models.

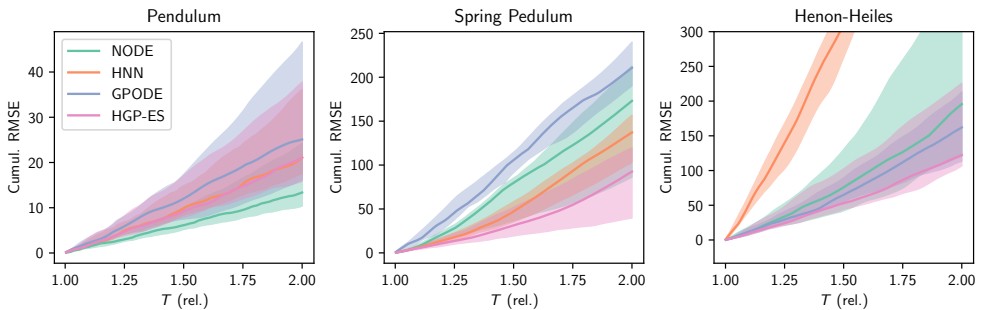

Figure 5: **The HGP with energy shooting has lowest cumulative trajectory error for the complex systems.** Cumulative error over test period $[T, 2T]$ for each system in the forecasting task. The horizontal axis shows the time relative to the start of the training period.

**Effect of trajectory length.** Figure 6a shows the effect of increasing trajectory length on model performance for task 1 on the HH system. The point-wise predictive performance of each model stays approximately constant with increasing trajectory length, with performances degrading slightly for the longest trajectories,

---

[1]We aim to give trajectory lengths that cover a similar amount of phase space for each system for fair comparison, with the different times reflecting the different properties of each system.

[2]Animations of the trajectories can be found at `https://magnusross.github.io/HamiltonianGPs/`.

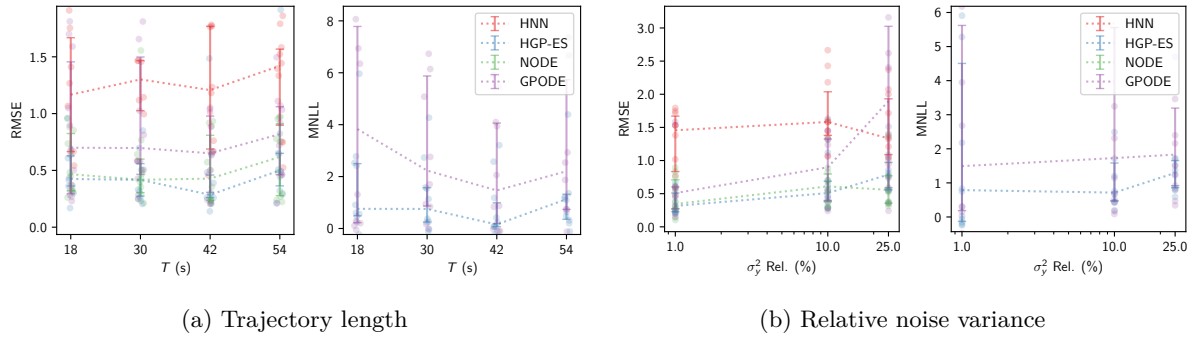

(a) Trajectory length

(b) Relative noise variance

Figure 6: **The HGP performance is robust to observation length and noise.** Analysis of effect of data for the HH system on task 1.

due to the difficulty of the optimisation problem. The HGP gives consistently good performance over each trajectory length relative to the baseline models.

**Effect of noise.** Figure 6b shows the effect of increasing noise on performance, with the horizontal axis showing the noise variance as a percentage of total signal variance. The results show that, in terms of RMSE, both the NODE and HGP are largely robust to increasing noise, although the is some drop, with the GPODE degrading more severely.

| Initialisation | RMSE ($\downarrow$) | MNLL ($\downarrow$) |
|---|---|---|
| Random | 1.03 (0.06) | 1.49 (0.09) |
| Hamiltonian | **0.36** (0.19) | **0.42** (0.83) |

Table 2: Effect of inducing initialisation.

**Effect of initialisation.** Table 2 illustrates the effect of the Hamiltonian aware initialisation scheme discussed in section 6.1. We ran HGP on task 1 for the HH system, with randomly initialised inducing mean, and our proposed initialisation. The models initialised randomly fail to fit the data well, and produce solutions that extrapolate poorly.

**Comparing inference schemes.** Figure 7a provides a comparison between different inference methods for task 1 on the HH system with varying trajectory lengths. We compare the HGP with: energy conserving shooting (HGP-ES), standard shooting with no additional energy constraint (HGP-S), no shooting (HGP), and with inference on sub-sequences formed into batches (HGP-Batched). With the HGP-Batched model, we aim to emulate the inference method used for the SSGP and SPGP, see appendix B.3 for details. The results show that the HGP-ES provides the best or equivalent performance over each trajectory length, with

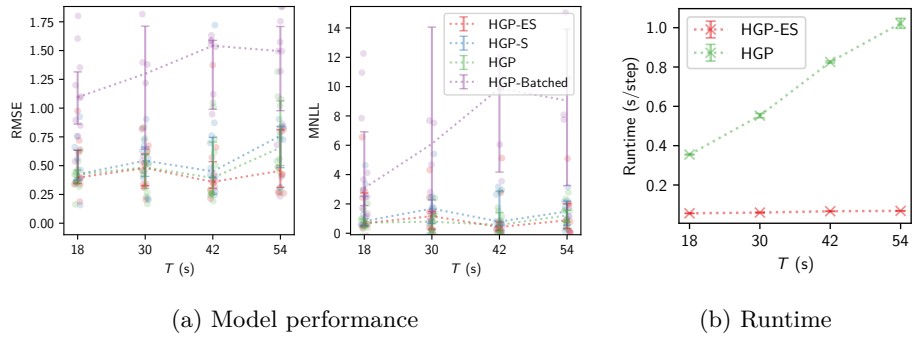

(a) Model performance

(b) Runtime

Figure 7: **Energy shooting inference performs best across trajectory lengths, whilst speeding up inference 15x for long trajectories.** Comparison of inference methods for task 1 on the HH system, 7a shows performance whist varying trajectory lengths, 7b compares the time taken to compute the evidence lower bound for the shooting and standard methods for varying trajectory lengths.

performance relative to alternative methods improving with increased trajectory length. The HGP-Batched method performs poorly for all trajectory lengths, with performance degrading for longer trajectories. Figure 7b shows the difference in runtimes for a single computation of the evidence lower for different trajectory lengths for the shooting and non-shooting models. The shooting approximation provides a $15\times$ speedup for the longest (54s) trajectory and $6.5\times$ speedup for the shortest (18s) trajectory, whilst matching or improving on the standard HGP in terms of RMSE and MNLL.

### 6.4 Task 2: Initial condition extrapolation

| | State RMSE ($\downarrow$) | | | State MNLL ($\downarrow$) | | | Energy RMSE ($\downarrow$) | | |
|---|---|---|---|---|---|---|---|---|---|
| Method | FP | HH | SP | FP | HH | SP | FP | HH | SP |
| NODE | **0.48** (0.15) | **1.15** (0.19) | **0.92** (0.38) | - | - | - | **0.26** (0.06) | **0.03** (0.00) | **1.35** (0.31) |
| HNN | 0.91 (0.47) | **1.35** (0.18) | 1.13 (0.26) | - | - | - | **0.25** (0.15) | 0.05 (0.03) | **0.96** (1.79) |
| GPODE | 2.41 (2.28) | 3.58 (0.79) | 3.53 (1.14) | 7.11 (8.22) | 19.11 (10.51) | 11.48 (10.73) | 6.17 (10.30) | 1.52 (0.95) | $1.9\times10^{7}(2.5\times10^{7})$ |
| HGP | 1.41 (0.46) | **1.30** (0.28) | 1.29 (0.59) | **4.68** (2.95) | **7.60** (3.03) | **6.13** (0.81) | 0.55 (0.60) | **0.03** (**0.01**) | 3.32 (8.86) |

Table 3: Performance comparison of different methods on the initial condition extrapolation task, with $K = 8$ trajectories.

Table 3 shows the results for each model on each systems for task 2. For this task each model was given a set of $K = 8$ noisy trajectories, of lengths 12, 4 and 6 seconds for the HH, FP and SP systems respectively, with initial conditions sampled randomly in phase space. The test set consists of 25 trajectories sampled from phase space using the same procedure, with length triple that of the training trajectories. In terms of state RMSE, the best performing model is the NODE, which provides significantly better predictions than the HGP for the simplest system (FP) and marginally better predictions for the more complex systems. The HGP performs better than the GPODE on all metrics, again illustrating advantages of the energy conserving prior. The HNN and NODE perform best in terms of energy RMSE, with HGP performing slightly worse, and the GPODE performing poorly.

**Number of trajectories** Figure 8 shows the performance of each model on the HH system for task 2 as the number of training trajectories $K$ is increased. We can see that whilst the HGP performs better than the GPODE on both metrics for all $K$, for large $K$ the NN models provide better point predictions. We believe the poor relative performance of the GP based models in task 2 can be attributed to the choice of prior, which we discuss in section 7.

Figure 8: **The HGP outperforms GPODE for all $K$, but NNs are best for large $K$.**

## 7 Discussion and limitations

**Performance of HNN vs NODE.** We found that in both tasks 1 and 2, the HNN model performed significantly worse than the NODE, which is unexpected and differs from previous results in the literature. We offer two possible explanations for this, both of which likely play a role. Firstly we use data that is both noisier, and use longer, more sparsely sampled trajectories than previous studies. Additionally in order to provide a fair comparison across methods, we learn $\mathcal{H}_{\boldsymbol{\theta}}(\mathbf{x})$, so that there is no restrictions on the set of Hamiltonians. For mechanical systems, the Hamiltonian can be rewritten in terms of the potential and kinetic energies, and a NN can be used to represent each term. Zhong et al. (2019) find that learning the mechanical Hamiltonian in this way leads to improved performance. We believe it should be possible to modify our framework to learn the potential and kinetic terms separately using a HGP, and this is an avenue we would like to pursue in future work.

**Suitability of the prior.** The performance of the HGP on task 2, is somewhat underwhelming especially for larger $K$, relative to NODE. We believe some of this effect can be attributed to our choice of GP prior.

We use a stationary GP prior, which is likely non-optimal for most Hamiltonian systems, which are typically nonstationary. The assumption of non-stationarity likely leads to the poor generalisation at new phase space points, since the GP prior will revert to zero mean functions there, which is not the case for the NN models. One option to improve the suitability of the GP prior is via the use of specifically designed kernels to represent symmetries in the system (Ridderbusch et al., 2021), or kernel structure learning (Kim & Teh, 2018). Another is to extend our framework and instead represent $\mathcal{H}$ with a deep GP prior (Damianou & Lawrence, 2013). Deep GPs are adept at modelling complex, nonstationary functions, and so would be well suited to the task. Using the SVI scheme proposed by Salimbeni & Deisenroth (2017) would make integration of deep GPs into our framework relatively straight forward, and would be interesting to undertake as part of future work.

**Control.** The present method has significant potential at improving Bayesian online (Deisenroth & Rasmussen, 2011) or policy-based RL (Yildiz et al., 2021) by incorporating Hamiltonian inductive biases. This requires expanding the model towards Hamiltonian systems with external forces, which relax the energy conservation assumption.

## 8   Conclusion

In this work we presented a Gaussian process model to learn Hamiltonian dynamical systems from trajectory observations. We proposed a parameterisation that combines inducing points and Fourier bases, and introduced a novel energy-conserving shooting method to allow reliable inference from long data. Our experiments show strong and stable performance under various learning settings.

### Acknowledgements

We would like to thank Pashupati Hegde for useful conversations and advice, and in particular for help with the code for the GPODE. We would also like to thank Mauricio Álvarez and Tom McDonald for feedback on earlier drafts of this work. This work was supported by MR's research visit to Aalto University through the ELLIS network. The calculations were performed using resources within the Aalto University Science-IT project. This work has been supported by the Academy of Finland (grant 334600).

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

# A  Variational bound derivations

## A.1  Standard bound derivation

We wish to minimise the KL $\arg\min_\theta \mathrm{KL}\left[q_\theta(\mathbf{u}, \mathbf{x}_0) \,||\, p(\mathbf{u}, \mathbf{x}_0|\mathbf{Y})\right]$ which corresponds to maximising the evidence lower bound $\log p(\mathbf{Y}) \geq \mathcal{F}(\theta, \mathbf{Z})$, with

$$\mathcal{F}(\theta, \mathbf{Z}) = \iint q(\mathbf{u}, \mathbf{x}_0) \log \frac{p(\mathbf{Y}, \mathbf{u}, \mathbf{x}_0)}{q(\mathbf{u}, \mathbf{x}_0)} d\mathbf{u} d\mathbf{x}_0 \tag{31}$$

$$= \iint q(\mathbf{u}, \mathbf{x}_0) \log p(\mathbf{Y}|\mathbf{u}, \mathbf{x}_0) \frac{p(\mathbf{u})}{q(\mathbf{u})} \frac{p(\mathbf{x}_0)}{q(\mathbf{x}_0)} d\mathbf{u} d\mathbf{x}_0 \tag{32}$$

$$= \underbrace{\iint q(\mathbf{u}, \mathbf{x}_0) \log p(\mathbf{Y}|\mathbf{u}, \mathbf{x}_0) d\mathbf{u} d\mathbf{x}_0}_{\mathcal{F}_{\mathbf{Y}}} + \underbrace{\int q(\mathbf{u}) \log \frac{p(\mathbf{u})}{q(\mathbf{u})} d\mathbf{u}}_{\mathcal{F}_{\mathbf{u}}} + \underbrace{\int q(\mathbf{x}_0) \log \frac{p(\mathbf{x}_0)}{q(\mathbf{x}_0)} d\mathbf{u}}_{\mathcal{F}_{\mathbf{x}_0}}. \tag{33}$$

The ELBO decomposes into three terms,

$$\mathcal{F} = \mathcal{F}_{\mathbf{Y}} + \mathcal{F}_{\mathbf{u}} + \mathcal{F}_{\mathbf{x}_0}. \tag{34}$$

The latter terms are KL divergences,

$$\mathcal{F}_{\mathbf{u}} = -\,\mathrm{KL}[q(\mathbf{u})||p(\mathbf{u})]$$
$$\mathcal{F}_{\mathbf{x}_0} = -\,\mathrm{KL}[q(\mathbf{x}_0)||p(\mathbf{x}_0)],$$

which can be computed in closed form, since we use Gaussian priors $p$ and variational approximations $q$ for both variables. Using the fact that

$$p(\mathbf{Y}|\mathbf{u}, \mathbf{x}_0) = \int p(\mathbf{Y}|\tilde{\mathbf{w}}, \mathbf{u}, \mathbf{x}_0) p(\tilde{\mathbf{w}}) d\tilde{\mathbf{w}} \tag{35}$$

$$= \mathbb{E}_{p(\tilde{\mathbf{w}})} \left[ p(\mathbf{Y}|\tilde{\mathbf{w}}, \mathbf{u}, \mathbf{x}_0) \right] \tag{36}$$

we can write

$$\mathcal{F}_{\mathbf{Y}} = \iint q(\mathbf{u}, \mathbf{x}_0) \log p(\mathbf{Y}|\mathbf{u}, \mathbf{x}_0) d\mathbf{u} d\mathbf{x}_0 \tag{37}$$

$$= \iint q(\mathbf{u}, \mathbf{x}_0) \log \mathbb{E}_{p(\tilde{\mathbf{w}})} \left[ p(\mathbf{Y}|\tilde{\mathbf{w}}, \mathbf{u}, \mathbf{x}_0) \right] d\mathbf{u} d\mathbf{x}_0 \tag{38}$$

$$\geq \mathbb{E}_{p(\tilde{\mathbf{w}})} \left[ \iint q(\mathbf{u}, \mathbf{x}_0) \log p(\mathbf{Y}|\tilde{\mathbf{w}}, \mathbf{u}, \mathbf{x}_0) d\mathbf{u} d\mathbf{x}_0 \right] \tag{39}$$

$$= \mathbb{E}_{p(\tilde{\mathbf{w}})} \mathbb{E}_{q(\mathbf{u})q(\mathbf{x}_0)} \log p(\mathbf{Y}|\tilde{\mathbf{w}}, \mathbf{u}, \mathbf{x}_0) \tag{40}$$

$$= \mathbb{E}_{p(\tilde{\mathbf{w}})q(\mathbf{u})q(\mathbf{x}_0)} \sum_{i=1}^{N} \log p(\mathbf{y}_i|\tilde{\mathbf{w}}, \mathbf{u}, \mathbf{x}_0), \tag{41}$$

where we applied Jensen's inequality in line (39). We can compute this term using Monte Carlo averaging.

## A.2 Shooting bound derivation

We wish to minimise the KL $\arg\min_\theta \text{KL}\left[q_\theta(\mathbf{u}, \mathbf{S}) \,||\, p(\mathbf{u}, \mathbf{S}|\mathbf{Y})\right]$ which corresponds to maximising the evidence lower bound $\log p(\mathbf{Y}) \geq \mathcal{F}(\theta, \mathbf{Z})$, with

$$\mathcal{F}(\theta, \mathbf{Z}) = \iint q(\mathbf{u}, \mathbf{S}) \log \frac{p(\mathbf{Y}, \mathbf{u}, \mathbf{S})}{q(\mathbf{u}, \mathbf{S})} d\mathbf{u} d\mathbf{S} \tag{42}$$

$$= \underbrace{\iint q(\mathbf{u}, \mathbf{S}) \log \mathbb{E}_{p(\tilde{\mathbf{w}})}\left[\prod_{i=1}^N p(\mathbf{y}_i|\tilde{\mathbf{w}}, \mathbf{u}, \mathbf{s}_{l(i)}) \prod_{l=1}^L p(\mathbf{s}_l|\mathbf{s}_{l-1}, \tilde{\mathbf{w}}, \mathbf{u})\right] d\mathbf{u} d\mathbf{S}}_{\mathcal{F}_{\mathbf{Ys}}} \tag{43}$$

$$\underbrace{- \int q(\mathbf{S}) \log \prod_{l=1}^L q(\mathbf{s}_l) d\mathbf{S}}_{\mathcal{F}_{\mathbf{s}}} + \underbrace{\int q(\mathbf{s}_0) \log \frac{p(\mathbf{s}_0)}{q(\mathbf{s}_0)} d\mathbf{s}_0}_{\mathcal{F}_{\mathbf{s}_0}} + \underbrace{\int q(\mathbf{u}) \log \frac{p(\mathbf{u})}{q(\mathbf{u})} d\mathbf{u}}_{\mathcal{F}_{\mathbf{u}}}, \tag{44}$$

where in the second line we have used the fact that,

$$p(\mathbf{Y}, \mathbf{u}, \mathbf{S}) = \int p(\mathbf{Y}, \mathbf{u}, \mathbf{S}, \tilde{\mathbf{w}}) d\tilde{\mathbf{w}} \tag{45}$$

$$= \int \prod_{i=1}^N p(\mathbf{y}_i|\tilde{\mathbf{w}}, \mathbf{u}, \mathbf{s}_{l(i)}) \prod_{l=1}^L p(\mathbf{s}_l|\mathbf{s}_{l-1}, \tilde{\mathbf{w}}, \mathbf{u}) p(\tilde{\mathbf{w}}) d\tilde{\mathbf{w}} p(\mathbf{u}) p(\mathbf{s}_0) \tag{46}$$

$$= \mathbb{E}_{p(\tilde{\mathbf{w}})}\left[\prod_{i=1}^N p(\mathbf{y}_i|\tilde{\mathbf{w}}, \mathbf{u}, \mathbf{s}_{l(i)}) \prod_{l=1}^L p(\mathbf{s}_l|\mathbf{s}_{l-1}, \tilde{\mathbf{w}}, \mathbf{u})\right] p(\mathbf{u}) p(\mathbf{s}_0). \tag{47}$$

The latter terms in the bound are given by,

$$\mathcal{F}_{\mathbf{s}} = \sum_{l=1}^L \mathbb{H}[q(\mathbf{s}_l)] \tag{48}$$

$$\mathcal{F}_{\mathbf{s}_0} = -\text{KL}[q(\mathbf{s}_0) \,||\, p(\mathbf{s}_0)] \tag{49}$$

$$\mathcal{F}_{\mathbf{u}} = -\text{KL}[q(\mathbf{u}) \,||\, p(\mathbf{u})] \tag{50}$$

where $\mathbb{H}$ represents the entropy, and each can be computed in closed form, since we again use Gaussian priors $p$ and variational approximations $q$. We can decompose the term $\mathcal{F}_{\mathbf{Ys}}$ further,

$$\mathcal{F}_{\mathbf{Ys}} = \iint q(\mathbf{u}, \mathbf{S}) \log \mathbb{E}_{p(\tilde{\mathbf{w}})}\left[\prod_{i=1}^N p(\mathbf{y}_i|\tilde{\mathbf{w}}, \mathbf{u}, \mathbf{s}_{l(i)}) \prod_{l=1}^L p(\mathbf{s}_l|\mathbf{s}_{l-1}, \tilde{\mathbf{w}}, \mathbf{u})\right] d\mathbf{u} d\mathbf{S} \tag{51}$$

$$\geq \mathbb{E}_{p(\tilde{\mathbf{w}})}\left[\iint q(\mathbf{u}, \mathbf{S}) \log \prod_{i=1}^N p(\mathbf{y}_i|\tilde{\mathbf{w}}, \mathbf{u}, \mathbf{s}_{l(i)}) \prod_{l=1}^L p(\mathbf{s}_l|\mathbf{s}_{l-1}, \tilde{\mathbf{w}}, \mathbf{u}) d\mathbf{u} d\mathbf{S}\right] \tag{52}$$

$$= \mathbb{E}_{p(\tilde{\mathbf{w}})}\left[\sum_{i=1}^N \iint q(\mathbf{u}, \mathbf{S}) \log p(\mathbf{y}_i|\tilde{\mathbf{w}}, \mathbf{u}, \mathbf{s}_{l(i)}) d\mathbf{u} d\mathbf{s}_{l(i)}\right] \tag{53}$$

$$+ \mathbb{E}_{p(\tilde{\mathbf{w}})}\left[\sum_{l=1}^L \iint \log p(\mathbf{s}_l|\mathbf{s}_{l-1}, \tilde{\mathbf{w}}, \mathbf{u}) d\mathbf{u} d\mathbf{s}_l d\mathbf{s}_{l-1}\right] \tag{54}$$

$$= \mathbb{E}_{p(\tilde{\mathbf{w}})q(\mathbf{u})}\left[\sum_{i=1}^N \mathbb{E}_{q(\mathbf{s}_{l(i)})}\left[\log p(\mathbf{y}_i|\tilde{\mathbf{w}}, \mathbf{u}, \mathbf{s}_{l(i)})\right] + \sum_{l=1}^L \mathbb{E}_{q(\mathbf{s}_l)q(\mathbf{s}_{l-1})}\left[\log p(\mathbf{s}_l|\mathbf{s}_{l-1}, \tilde{\mathbf{w}}, \mathbf{u})\right]\right]. \tag{55}$$

where we applied Jensen's inequality in line (52). We can compute this term using Monte Carlo averaging. Together the bound is given by,

$$\mathcal{F} = \mathcal{F}_{\mathbf{Ys}} + \mathcal{F}_{\mathbf{s}} + \mathcal{F}_{\mathbf{s}_0} + \mathcal{F}_{\mathbf{u}}. \tag{56}$$

### A.3 Multiple trajectory variational bounds

To extend the bounds given in Equations (24) and (30) we consider a set of $K$ trajectories, $\mathbf{Y} = \{\mathbf{Y}_1, \ldots, \mathbf{Y}_K\}$, for ease of notation we assume each trajectory has the same number of points $N$, and the same observation times $t_i \in (t_1, \ldots, t_N)$, but this is not a limitation of the model itself. Each trajectory will have distinct initial conditions, so we aim to infer the set $\mathbf{X}_0 = \{\mathbf{x}_{1,0}, \ldots, \mathbf{x}_{K,0}\}$. We assume the variational posterior over each initial state factorises, so

$$q(\mathbf{X}_0) = \prod_{k=1}^{K} q(\mathbf{x}_{k,0}) = \prod_{k=1}^{K} \mathcal{N}(\mathbf{x}_{k,0}|\mathbf{m}_{k,0}, \mathbf{Q}_{k,0}).$$

Given this factorisation, for the standard HGP model without the shooting approximation, we obtain the bound

$$\mathcal{F}(\theta, \mathbf{Z}) = \mathbb{E}_{q(\mathbf{u})q(\mathbf{X}_0)p(\tilde{\mathbf{w}})} \left[ \sum_{k=1}^{K} \sum_{i=1}^{N} \log p(\mathbf{y}_{k,i}|\tilde{\mathbf{w}}, \mathbf{u}, \mathbf{x}_{k,0}) \right] - \mathrm{KL}\left[q(\mathbf{u})||p(\mathbf{u})\right] - \sum_{k=1}^{K} \mathrm{KL}\left[q(\mathbf{x}_{k,0})||p(\mathbf{x}_{k,0})\right]. \quad (57)$$

where $\theta = (\mathbf{m}, \mathbf{Q}, \{\mathbf{m}_{k,0}, \mathbf{Q}_{k,0}\})$. To extented the the shooting HGP to multiple trajectories, we require a set of shooting states for each trajectory, and aim to infer $\mathbf{S} = \{\mathbf{S}_1, \ldots, \mathbf{S}_K\}$, where we assume that each trajectory has the same number of shooting states $L$. We again assume that the variational posterior over the shooting states factorises, so

$$q(\mathbf{S}) = \prod_{k=1}^{K} q(\mathbf{S}_k) = \prod_{k=1}^{K} \prod_{l=0}^{L} \mathcal{N}(\mathbf{s}_{k,l}|\mathbf{a}_{k,l}, \Sigma_{k,l}),$$

which leads to the bound

$$\mathcal{F}(\theta, \mathbf{Z}) = \mathbb{E}_{q(\mathbf{u})p(\tilde{\mathbf{w}})} \Big[ \sum_{k=1}^{K} \sum_{i=1}^{N} \mathbb{E}_{q(\mathbf{s}_{k,l(i)})} \left[ \log p(\mathbf{y}_i|\tilde{\mathbf{w}}, \mathbf{u}, \mathbf{s}_{k,l(i)}) \right] \quad (58)$$

$$+ \sum_{k=1}^{K} \sum_{l=0}^{L} \mathbb{E}_{q(\mathbf{s}_{k,l})q(\mathbf{s}_{k,l-1})} \left[ \log p(\mathbf{s}_{k,l}|\mathbf{s}_{k,l-1}, \tilde{\mathbf{w}}, \mathbf{u}) \right] \Big] \quad (59)$$

$$- \sum_{k=1}^{K} \sum_{l=1}^{L} \mathbb{H}[q(\mathbf{s}_{k,l})] - \sum_{k=1}^{K} \mathrm{KL}[q(\mathbf{s}_{k,0}) \,||\, p(\mathbf{s}_{k,0})] - \mathrm{KL}[q(\mathbf{u}) \,||\, p(\mathbf{u})],$$

where $\theta = (\mathbf{m}, \mathbf{Q}, \{\mathbf{a}_{k,l}, \Sigma_{k,l}\})$, and $\mathbf{s}_{k,l(i)}$ represents the last shooting state before time $t_i$ for trajectory $k$.

### A.4 Model Complexities

The dominant complexity comes from the sampling of inducing points, which is $\mathcal{O}(M^3)$ due to the requirement to compute the Cholesky decomposition of the covariance matrix. The number of evaluations of the derivative function is approximately proportional to the length of the training period $T$, meaning that for a $D$ dimensional system the complexity is $\mathcal{O}(DT)$, due to the decoupled sampling of GPs being linear with respect to the number of inputs. For the shooting model, integration over $T$ can be parallelised, leading to a lower runtime. Large systems are likely to be more complex, and require more inducing points. Therefore, the complexity with respect to the number of inducing points provides some upper limit on the system size for which we can obtain good performance, although it is typically possible to use 1000s of inducing points, which would correspond to very complex Hamiltonians.

## B Experiments

### B.1 Hamiltonian systems

This section gives further details on the systems used for the experiments. For all systems, we scale the data so the training points have zero mean and unit variance, and compute metrics on the scaled data, so they are comparable across systems.

**Fixed Pendulum**   The fixed pendulum Hamiltonian represents the motion of a frictionless pendulum rotating about a single axis, under the influence of gravity, with the coordinate $q$ representing the angle of the pendulum to the vertical, and the coordinate $p$ representing the corresponding angular momentum. We set $m = 1$kg, $r = 1$m and $g = 9.81$m/s. We sample initial conditions by drawing $p, q \sim U(-1, 1)$, and discard those with energy $E > mgr$ to avoid the pendulum swinging over its pivot. We use a sampling rate of 8Hz for the training data and 15Hz for the test data.

**Spring pendulum**   The spring pendulum system (Lynch, 2000) is similar to the fixed pendulum, but additionally models the pendulum shaft as a Hookian spring, which can extend and contract as forces act upon it. In this system, $q_1$ represents the angle of the pendulum to the vertical, $q_2$ the extension of the shaft relative to its resting position, and $p_1$ and $p_2$ are the corresponding angular and linear momentum respectively. We set $m = 1$kg, $r = 3$m and $g = 9.81$m/s, we set the spring constant to $k = 10$N/m. We sample initial conditions as $q_1, q_2, p_1, p_2 \sim U(-0.25, 0.25)$. We use a sampling rate of 6Hz for the training data, and 10Hz for the test data.

**Henon-Heiles system**   The Henon-Heiles Hamiltonian represents the motion of a star around a galactic centre, with $q_1$ and $q_2$ representing spatial coordinates in the plane and $p_1$ and $p_2$ the corresponding linear momenta (Henon & Heiles, 1964). We set $\mu = 0.8$m$^{-1}$. We sample initial conditions by first sampling $q_1, q_2, p_1, p_2 \sim U(-1, 1)$ and then discarding initial conditions with $E > \frac{1}{6\mu^2}$ to ensure trajectories stay in the region of phase space with connected level sets, that is to say the trajectories stay localised after long times, and do not tend to infinity (Offen & Ober-Blöbaum, 2022). We use a sampling rate of 4Hz for the training data, and 10Hz for the test data.

### B.2   Hamiltonian aware initialisation

We form approximate data for the system's derivative function by computing the numerical derivatives of the trajectory data. For example using the first differences approximation,[3] we obtain $\dot{\mathbf{Y}} = (\frac{\mathbf{y}_2 - \mathbf{y}_1}{t_2 - t_1}, \frac{\mathbf{y}_3 - \mathbf{y}_2}{t_3 - t_2}, ..., \frac{\mathbf{y}_N - \mathbf{y}_{N-1}}{t_N - t_{N-1}}) \in \mathbb{R}^{N \times 2D}$. We treat $\dot{\mathbf{Y}}$ as observed data for the derivative function $\mathbf{f}$ at input locations $\mathbf{Y} \in \mathbb{R}^{N \times 2D}$. We wish to obtain the mean of the Hamiltonian implied by this data, at the inducing input locations $\mathbf{Z}$, which we can use to initialise the variational distributions for the inducing variables. Let $\mathrm{vec}(\dot{\mathbf{Y}}) \in \mathbb{R}^{2ND}$ denote a 'flattened' $\dot{\mathbf{Y}}$, then this mean is given by

$$\mathbf{m} = \mathbf{k}_{\mathcal{H}\mathbf{f}}(\mathbf{Z}, \mathbf{Y}) \mathbf{K}_{\mathbf{f}}(\mathbf{Y}, \mathbf{Y}) \, \mathrm{vec}(\dot{\mathbf{Y}}) \tag{60}$$

where $\mathbf{k}_{\mathcal{H}\mathbf{f}}(\mathbf{Z}, \mathbf{Y}) \in \mathbb{R}^{M \times 2ND}$ is the matrix from by evaluating the covariance in equation (11) for each pair of input points, and similarly $\mathbf{K}_{\mathbf{f}}(\mathbf{Y}, \mathbf{Y}) \in \mathbb{R}^{2ND \times 2ND}$ is the covariance in equation (12) evaluated at each pair of input points.

### B.3   Models

This section provides further details on the model setup.

**HGP**   For the HGP model we use the ARD kernel. We use a whitened representation of the inducing variables $\mathbf{u}$ for optimisation. We optimise the variational bound with respect to the variational parameters $\theta$, and the various hyperparameters of the model: $\sigma_{\mathrm{obs}}^2$ the noise variance, the kernel lengthscales and signal variance, and the inducing input locations. We fix the shooting constraint variance as $\sigma_\xi^2 = 1 \times 10^{-6}$, and the energy constraint variance to $\sigma_\chi^2 = 2.5 \times 10^{-3}$.

**GPODE**   For the GPODE we mirror the setup of the HGP where possible. We follow Hegde et al. (2022) and place independent GP priors over the each component of the derivative function, each with an ARD kernel and a distinct set of inducing variables. We optimise the variational bound with respect to the same set of variational parameters and model hyperparameters. We use the same initialisation process for the inducing variational means as described by Hegde et al. (2022)

---

[3]In practice we estimate derivatives using the `gradient` function, from the `numpy` package(Harris et al., 2020).

**NN models** For both the HNN and the NODE we use 3 hidden layers of size 256, with tanh activation. To train the NN models we form a set of sub-sequences by sliding a window of length 6 samples over the data, moving the window 1 sample at a time over the training range, e.g. for a single trajectory of length 100, we obtain $100 - 6 = 94$ sub-sequences. We take these sub-sequences, shuffle them, and form them into batches. We train the model by rolling out the NN parameterised derivative function over each sub-sequence in the batch, using the first value as the initial condition, and optimising the L1 loss with respect to the training data. We use the Adam optimiser with learning rate $3 \times 10^{-3}$. Because the model is trained on short sub-sequences, and is unaware that continuity is required between sub-sequences, it is liable to overfit and produce a solution that performs poorly when integrated over the entire long trajectory. To avoid this, every 10 epochs we integrate over the entire trajectory from the first point in the training data and compute the loss, we select the model with the best full trajectory loss after training has proceeded for a fixed number of iterations. For the experiments on task 1 we use a batch size of 16 and for those on task 2 we use a batch size of 32.

**HGP-Batched in relation to SSGP/SPGP** To compare the our energy conserving shooting method with the training method used in the SSGP and SPGP we re-implement the training method based on short sub-sequences for our HGP model. The SSGP and SPGP use slightly different schemes, but both use a method similar to that described in the proceeding paragraph. Short sub-sequences are formed from the full, long trajectories by sliding a fixed size window over the data. We again shuffle the sub-sequences and form them into batches. In training, we roll out the model and compute the bound over the short sub sequences in each batch. This avoids the problem of vanishing/exploding gradients, but means we are optimising the model on a related but different task. A model can perform well on the short sub trajectories, but poorly when rolled out over the full trajectory. To avoid overfitting we roll out the model over the full trajectory after a set number of epochs, and compute the MNLL, selecting the model with the lowest value after a fixed number of training iterations. For the HGP-Batched model we used a sub sequence size of 6, and a batch size of 16 for all experiments. Additionally we follow (Ensinger et al., 2022) and do not learn initial conditions, but instead use the first data point in each sequence as its initial condition. It should be noted that Ensinger et al. (2022) use a batch size of 1 and different sequence lengths (10-50) for each problem under consideration. We found that a larger batch size and a shorter sequence length performed better for our experiments. The procedure for the SSGP differs slightly because Tanaka et al. (2022) only consider task 2, that is to say learning from multiple trajectories. Instead of rolling out over the full training trajectory for validation, the authors keep a set of hold out trajectories, which the use to avoid overfitting, this is of course not applicable in the case of task 1. Additionally Tanaka et al. (2022) a sub sequence length equivalent to 1s of data, which means a different number of points for different systems depending on the sampling rate. For our implementation of the HGP-Batched model, we attempted to capture the key components of the methods proposed in the SSGP and the SPGP, and in order to make the comparison as fair as possible chose settings that provided the best performance on our experiments.

**Symplectic integration** Symplectic integrators specifically designed for Hamiltonian systems respect the conservation of the Hamiltonian during forward integration (Ensinger et al., 2022; Rath et al., 2021). This can lead to improved integration accuracy over long time scales. In our experiments we chose to use the non-symplectic Runge-Kutta 4/5 method (Dormand & Prince, 1980), due to the implementation being easily available in PyTorch. Our framework does not make any restrictions on the type of solver, and one could chose to use a symplectic solver if an implementation were available. It should be noted that for the HGP with shooting the choice of solver has very little effect on the solution learned in training, since integration is only happening over the very short shooting segments, which means the energy loss within a segment is negligible, and so a symplectic solver would be of little benefit to the learning algorithm.

### B.4 Implementation and hardware

All experiments were run on a MacBook Pro (14-inch, 2021) laptop with M1 Pro chip and 32 GB memory, using the CPU. We implement the HGP, and baseline models in Python, with the PyTorch framework (Paszke et al., 2019). We base our implementation on that of Hegde et al. (2022), which is available at

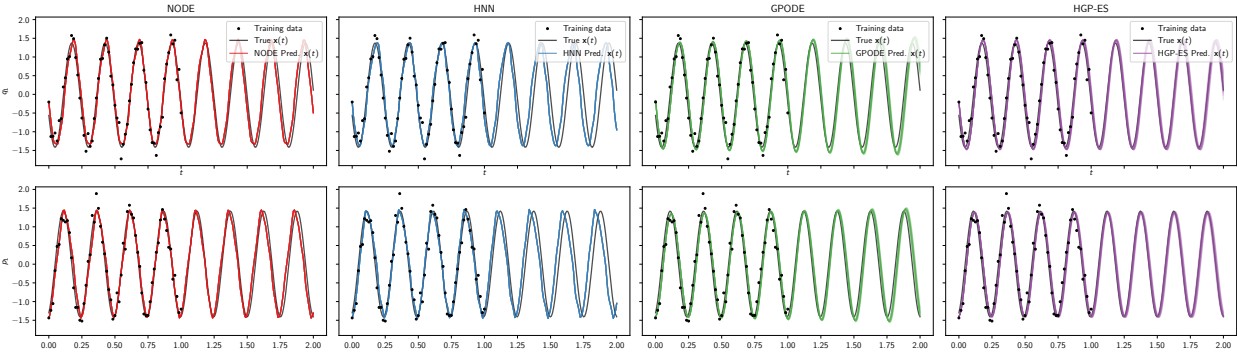

Figure 9: A sample trajectory showing training and test data, and model predictions for each model for a single repeat of task 1 on the FP system. For the GPODE and HGP, 32 samples from the model are shown.

`https://github.com/hegdepashupati/gaussian-process-odes`. We will on releasing the code along with the camera ready version of the paper if accepted.

## C Additional results

### C.1 Trajectory plots

Figures 9, 10 and 11 show the data and model predictions for the FP, SP and HH systems respectively, for one repeat of the results shown in Table 1. These plots are additionally shown in animation form in the supplementary material. Note that the animations show the noise free data for the training period, not the noisy data shown in the plots.

### C.2 Mean results tables

Tables 4 and 5 show the equivalent of tables 1 and 3 but with means and standard errors, as opposed to the median and interquartile range shown in the main text.

| Method | State RMSE (↓) | | | State MNLL (↓) | | | Energy RMSE (↓) | | |
| --- | --- | --- | --- | --- | --- | --- | --- | --- | --- |
| | FP | HH | SP | FP | HH | SP | FP | HH | SP |
| NODE | 0.16 (0.03) | 0.48 (0.08) | 1.04 (0.19) | - | - | - | 0.33 (0.08) | 0.02 (0.00) | 1.67 (0.72) |
| HNN | 0.38 (0.14) | 1.49 (0.11) | 1.40 (0.06) | - | - | - | 0.39 (0.09) | 0.07 (0.02) | 1.05 (0.11) |
| GPODE | 0.32 (0.08) | 0.65 (0.13) | 1.06 (0.22) | 0.77 (0.69) | 2.86 (1.11) | 4.31 (1.54) | 0.37 (0.09) | 0.02 (0.00) | 3.34 (1.78) |
| HGP | 0.23 (0.04) | 0.50 (0.11) | 0.56 (0.09) | 0.04 (0.20) | 1.63 (0.89) | 1.64 (0.55) | 0.31 (0.08) | 0.01 (0.00) | 0.80 (0.08) |

Table 4: Performance comparison of different methods on each system on the trajectory forecasting task. Table shows mean and standard error over 10 repeats, for the same data as in Table 1.

| Method | State RMSE (↓) | | | State MNLL (↓) | | | Energy RMSE (↓) | | |
| --- | --- | --- | --- | --- | --- | --- | --- | --- | --- |
| | FP | HH | SP | FP | HH | SP | FP | HH | SP |
| NODE | 0.47 (0.03) | 1.18 (0.04) | 1.05 (0.13) | - | - | - | 0.27 (0.03) | 0.03 (0.00) | 1.26 (0.18) |
| HNN | 0.88 (0.12) | 1.42 (0.12) | 1.20 (0.07) | - | - | - | 0.43 (0.19) | 0.02 (0.01) | 1.53 (0.73) |
| GPODE | 2.48 (0.41) | 3.50 (0.26) | 3.38 (0.27) | 8.56 (1.76) | 18.38 (2.55) | 11.07 (1.89) | 4.67 (2.21) | 1.42 (0.44) | $3.1 \times 10^7$ $(3.0 \times 10^7)$ |
| HGP | 1.33 (0.13) | 1.39 (0.13) | 1.49 (0.11) | 5.83 (1.23) | 7.96 (0.69) | 6.15 (0.34) | 0.80 (0.41) | 0.02 (0.00) | 6.06 (2.52) |

Table 5: Performance comparison of different methods on the initial condition extrapolation task, with $K = 8$ trajectories. Table shows mean and standard error over 10 repeats, for the same data as in Table 3.

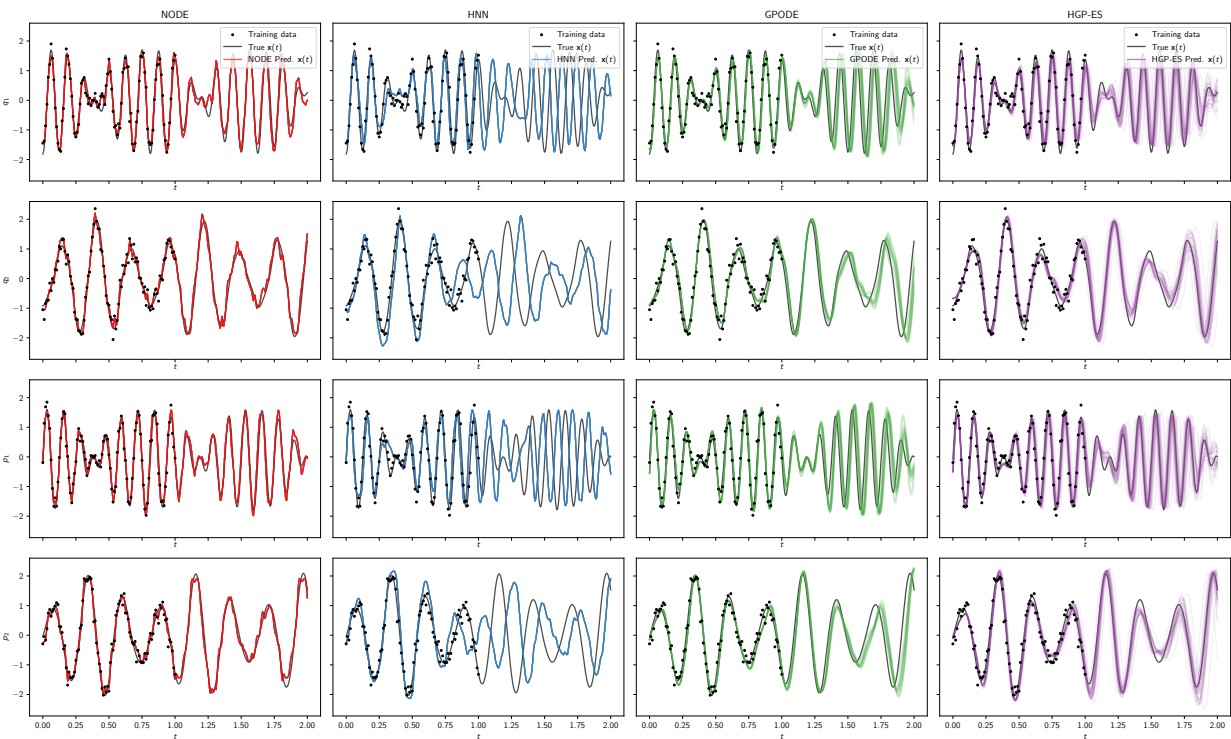

Figure 10: A sample trajectory showing training and test data, and model predictions for each model for a single repeat of task 1 on the SP system. For the GPODE and HGP, 32 samples from the model are shown.

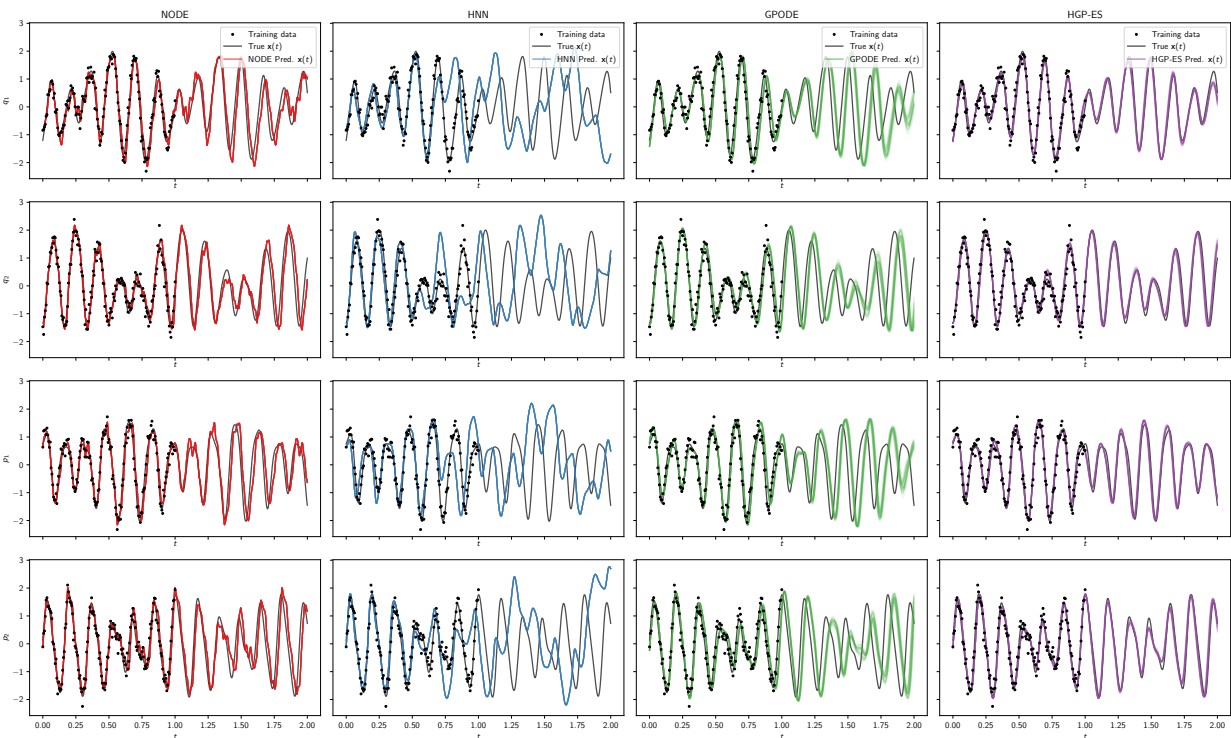

Figure 11: A sample trajectory showing training and test data, and model predictions for each model for a single repeat of task 1 on the HH system. For the GPODE and HGP, 32 samples from the model are shown.

## C.3 Comparison with SSGP

In this section we provide a comparsion with the SSGP model of Tanaka et al. (2022). The implementation of the SSGP provide by the authors only works for systems with $D = 1$, and is non-trivial to reimplement, so we are only able to obtain comparisons on the toy FP system. For the SSGP we used the same settings as Tanaka et al. (2022) throughout.

| Method | State RMSE (↓) | State MNLL (↓) | Energy RMSE (↓) |
|---|---|---|---|
| NODE | **0.12** (0.12) | - | **0.30** (0.28) |
| HNN | 0.20 (0.16) | - | **0.34** (0.33) |
| GPODE | 0.23 (0.26) | -0.03 (1.21) | **0.38** (0.34) |
| HGP-ES | 0.18 (0.19) | **-0.21** (0.67) | **0.25** (0.42) |
| SSGP | 1.49 (0.39) | 1.50 (0.20) | **0.23** (0.32) |

Table 6: Performance comparison with the SSGP on the trajectory forecasting task for the FP system.

Table 6 shows the results of the SSGP on task 1. As expected, the SSGP model performs poorly on the trajectory forecasting task, which mirrors our findings in our comparison with the HGP-Batched model, discussed in section 6.3. In terms of both RMSE and MNLL performance the HGP produces considerable better results. On this task the best model is the NODE, which gives slightly better results than the HGP.

| Method | State RMSE (↓) | State MNLL (↓) | Energy RMSE (↓) |
|---|---|---|---|
| NODE | **0.48** (0.15) | - | 0.26 (0.06) |
| HNN | 0.91 (0.47) | - | 0.25 (0.15) |
| GPODE | 2.41 (2.28) | 7.11 (8.22) | 6.17 (10.30) |
| HGP-ES | 1.41 (0.46) | 4.68 (2.95) | 0.55 (0.60) |
| HGP-Batched | **0.54** (0.31) | **0.66** (0.51) | **0.10** (0.03) |
| SSGP | 0.75 (0.50) | 0.89 (0.37) | **0.13** (0.03) |

Table 7: Performance comparison with the SSGP on the initial condition extrapolation task with $K = 8$ trajectories for the FP system.

Table 7 shows the results of the SSGP on task 2. The SSGP model performs well for task 2 in comparison to the HGP-ES, which is perhaps not surprising given that this is the task that SSGP focused on ($D = 1$, multiple trajectories). The SSGP performs considerably worse than NODE in terms of state RMSE, although in terms of energy RMSE it provides a good result. In order to determine the cause of the performance difference between SSGP and HGP-ES we also ran the HGP model with batching. We found the HGP-Batched provided better performance than the SSGP and HGP-ES on this task. This indicates that the performance difference between the HGP-ES and the SSGP on task 2 can be attributed to the use of batching as opposed to shooting inference, and implies that the other differences between the models (inference via inducing points vs RFFs, inferring the initial conditions, Hamiltonian aware initialisation) have a positive effect on performance, since the HGP-batched provides better results than the SSGP

Overall these results show that for systems with $D = 1$, the HGP with shooting provides significantly better trajectory forecasting performance, whereas the batching-based inference provides better extrapolation to new initial conditions, from a given set of initial conditions. The results for task 2 show that shooting based inference is likely non-optimal for the problem of trajectory extrapolation with multiple trajectories. Fitting multiple trajectories jointly is a more difficult optimisation problem, since the inducing points must be effectively distributed over a larger area of phase space. Adding the shooting objectives for this problem narrows down the space of feasible solutions and makes the optimisation problem more difficult, making it harder to find a good solution. We plan to investigate this effect further as part of future work.

## C.4 Investigating FP performance

On the FP system the HGP performs worse than the neural network baseline on both tasks 1 and 2, while this does not hold for systems HH and SP. To investigate this phenomenon, we run extended experiments of

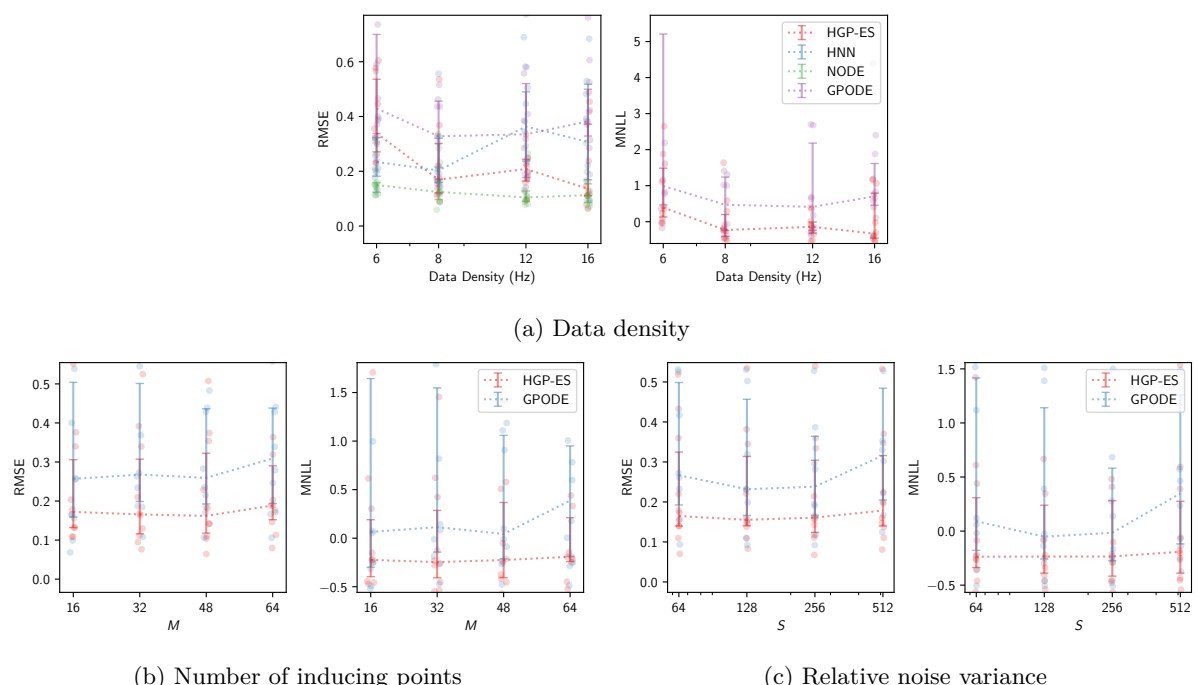

(a) Data density

(b) Number of inducing points        (c) Relative noise variance

Figure 12: Effect of different variables on model performance for the FP system on task 1.

the FP system to see if the performance of the HGP is improved. We vary (i) data density, (ii) number of inducing points, and (iii) number of random Fourier basis functions.

The results for task 1 are shown in Figure 12. We can see that increasing the data density (12a) provides a small improvement in the results for both the NODE and HGP, and seems to have a little effect on performance for the HNN and GPODE. Increasing the number of inducing points (12b) and basis functions (12c) does not effect performance significantly for the HGP.

The results for task 2 are shown in Figure 13. Figure 13a shows that increasing the data density for task 2 has a small positive effect on RMSE performance for the GP based models, and no significant effect for the NN based models. The effect on MNLL performance is not significant. Increasing the number of inducing points and basis functions again does not have a noticeable effect on the performance for the HGP.

These results do not suggest the density, number of basis functions, or number of inducing points is the cause for the poor relative performance of the GP models on task 2. We believe that the poor performance on task 2 is caused by the shooting bound being non optimal in the case of multiple trajectory inference as discussed in Section C.3.

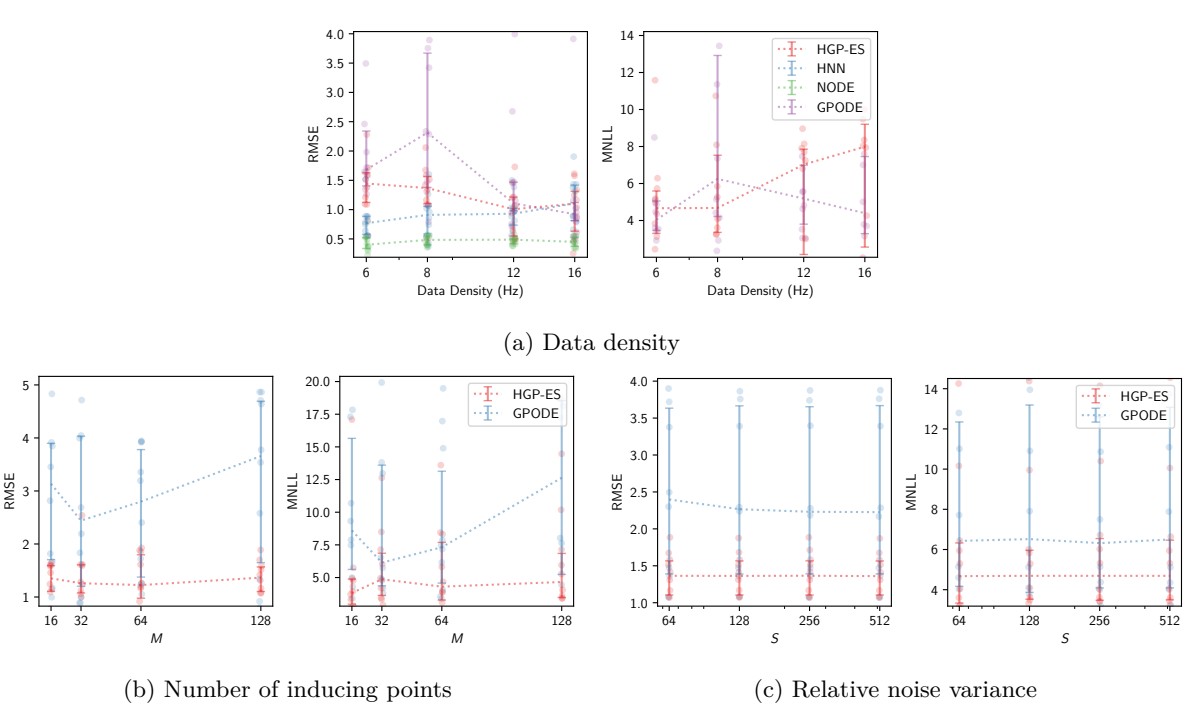

(a) Data density

(b) Number of inducing points      (c) Relative noise variance

Figure 13: Effect of different variables on model performance for the FP system on task 2.

