# OpenReview forum: "Learning Energy Conserving Dynamics Efficiently with Hamiltonian Gaussian Processes"
_TMLR — Accepted by TMLR_

### Review · Reviewer_14sk · 2022-12-20

**Summary Of Contributions:**

This paper extends [Hegde et al., 2022] to handle Hamiltonian dynamics. The Hamiltonian is modeled by the decoupled representation of Gaussian processes, which allows one to efficiently sample the Hamiltonian vector fields from the approximated GP posterior. The model learning is based on variational inference with multiple shooting schemes. This scheme can be used for effectively learning dynamics from long trajectories. The experiments on several physical systems show that the proposed method outperform the baselines (i.e., NODE, HNN, GPODE) for the HH and SP systems.

**Audience:**

Yes

**Claims And Evidence:**

Yes

**Requested Changes:**

- Compared to standard RFF, decoupling sampling has the advantage of being able to properly evaluate predictive variance. Can this be demonstrated in an experiment? Is it possible to visualize and analyze the variance in phase space, for example?

- HNN uses a learning method based on finite differences, so a comparison with SymODEN, which can be trained using ODE solver, would be appropriate. Would it be possible to add a comparison with SymODEN?

- I could not find the reason for the poor performance of HGP in the experiments on Pendulum. Would it be possible to analyze the results of the proposed method from the following perspectives.
For example,
  - What happens to the performance when the temporal resolution of the trajectory is low?
  - Is the number of inducing points and basis functions sufficient?

- Can the proposed method be applied to large systems? It would be good to add a discussion on the computational cost in that case.

- A learning method using symplectic integrators has been proposed to make energy conservation more rigorous, and I think it would be good to mention that and discuss whether the proposed method can take advantage of it.

- It is also good to mention whether it can be applied to systems with friction.

- I am having trouble understanding the relationship between the variables s,y,x. Please clarify.

- Please explain in detail how to sample the initial conditions when generating data in an experiment. Do you limit the energy range, etc.?

**Strengths And Weaknesses:**

S1. To my knowledge, this is the first example of modeling a Hamiltonian using decoupled sampling.

S2. The variational inference algorithm with multiple shooting is practically useful.

S3. The evaluation was conducted for two tasks, i.e., forecasting and simulation.

S4. The manuscript is readable.

W1. Experiments have not shown the benefits of using decoupled sampling.

W2. There are several unclear points regarding the experimental setup.

W3. The advantage of the proposed method is unclear.

W4. There is insufficient discussion of the advantages and adaptability of the proposed method.

---

> ### Author Response · Authors · 2022-12-21
> **Acknowledging review**
>
> We thank you for your review, and especially for the fast turnaround at this time of year. We would like to let you know that we have seen the review, but due to previous commitments over the holiday period, we will not be able to respond until approximately the 13th of January. This should still be well within the prescribed timeframe (4 weeks after all 3 reviews posted).  Thanks.

---

> ### Author Response · Authors · 2023-01-13
> **Response to Reviewer 14sk: Part 1**
>
> We thank the reviewer for their detailed review and appreciate the quick turnaround at a busy time in the year. Below, we address all the reviewer's concerns.
>
> > Compared to standard RFF, decoupling sampling has the advantage of being able to properly evaluate predictive variance. Can this be demonstrated in an experiment? Is it possible to visualize and analyze the variance in phase space, for example?
>
> While we agree that a direct comparison between the decoupled sampling and standard RFFs for our experiments would be interesting, it is difficult to directly compare them within our framework. In RFF we infer posterior over Fourier weights, while in our framework we infer over energy inducing points. The variational bounds (ELBOs), and large part of the implementation of the model, will be different in the two cases. One example of a difference is in initialisation, where we are able to use the Hamiltonian-aware initialisation described in Section 6.1 due to the interpretability of our inducing points, which would not be possible using RFFs. The results in (Wilson et al., 2020) give much experimental and theoretical evidence that uncertainty is better captured by decoupled sampling, and there is no reason that this should not translate to the HGP context, however reimplementing the comparison it in our framework properly is unfortunately not feasible in the time frame of the rebuttal period.
>
> > HNN uses a learning method based on finite differences, so a comparison with SymODEN, which can be trained using ODE solver, would be appropriate. Would it be possible to add a comparison with SymODEN?
>
> In the experiments the model labelled as HNN is a version of SymODEN, that is to say it represents the Hamiltonian with a NN and uses an ODE solver to compute trajectory rollouts, not using finite differences, this model is referred to as Unstructured SymODEN in the work of Zhong et al. (2019).  We thought it simpler to use the label HNN as this is what this model is referred to in some other work in the area, but we appreciate this is confusing since it is different to the original HNN model. This point was briefly discussed in Section 6.1 Paragraph "NN baselines", but we can add further clarification if necessary. We apologise for being unclear on this point.
>
> > I could not find the reason for the poor performance of HGP in the experiments on Pendulum. Would it be possible to analyze the results of the proposed method from the following perspectives. For example,
> > -   What happens to the performance when the temporal resolution of the trajectory is low?
> > -   Is the number of inducing points and basis functions sufficient?
>
> On Task 1 for the pendulum system, although NODE does slightly better, all models perform similarly well and provide good solutions (see Figure 9 in Appendix for plots of the trajectories). On Task 2  the HGP does significantly worse than the NODE/HNN, and as pointed out by the reviewer, performs poorly. In order to investigate the performance of the HGP on both task 1 and 2 for the pendulum systems, we ran the three experiments suggested by the reviewer for each task, varying:
> 1) The data density (6Hz, 8Hz, 12Hz, 16Hz).
> 2) The number of inducing points in the GP based models (16, 32, 48, 64 for task 1 and 16, 32, 64, 128 for task 2.
> 3) The number of basis functions in the GP based models (64, 128, 256, 512).
>
> The six plots showing the results of these experiments can be found in the updated manuscript, in section C.4.  We give a short summary of the findings below:
> - For task 1, increasing the data density provides a small improvement in the results for both the NODE and HGP, and seems to have a little effect on performance for the HNN and GPODE.
> - For task 2, increasing the data density improves the performance of the GPODE and HGP relative to the NN models, with the performance of the HNN degrading and the NODE staying approximately the same.
> - Increasing $M$, the number of inducing points, seems to have a slightly negative effect on performance for the GPODE, and no effect on the HGP for both tasks.
> - Increasing $S$, the number of basis functions, seems to have no effect on either the HGP or the GPODE for either task.
>
> These results do not suggest the density, number of basis functions, or number of inducing points is the cause for the poor relative performance of the GP models on task 2. We believe that the poor performance is caused by the fact that fitting multiple trajectories jointly is a more difficult optimisation problem, since the inducing points must be effectively distributed over a larger area of phase space. The additional the shooting objectives for this problem narrows down the space of feasible solutions and makes the optimisation problem more difficult, making it harder to find a good solution. We plan to investigate this effect further as part of future work.

---

> ### Author Response · Authors · 2023-01-13
> **Response to Reviewer 14sk: Part 2**
>
> > Can the proposed method be applied to large systems? It would be good to add a discussion on the computational cost in that case.
>
> Yes, the proposed method should be applicable to large systems, since the complexity is $\mathcal{O}(D)$ with respect to the output dimension, we have added a clarification about this to the updated manuscript, in section A.4.  One point to note, large systems are likely to be more complex, and so require more inducing points to fully characterise given their high dimensional nature. The complexity with respect to the number of inducing points is $\mathcal{O}(M^3)$, so this does provides some upper limit on the system size for which we can obtain good performance, although it is typically possible to use 1000s of inducing points, which would correspond to a very complex Hamiltonian.
>
> > A learning method using symplectic integrators has been proposed to make energy conservation more rigorous, and I think it would be good to mention that and discuss whether the proposed method can take advantage of it.
>
> For our experiments we use the `dopri5` from `torchdiffeq` which is not symplectic, however our framework places no restrictions on the solver, any can be used. We chose to use that method, as no symplectic solvers are implemented in `torchdiffeq`, however it would, of course, be possible to add one and use it in our model. For the HGP with shooting the choice of solver has very little effect on the solution learned in training, since integration is only happening over the very short shooting segments, which means the energy loss within a segment is negligible, and so a symplectic solver would give little benefit. We have added a short discussion of this point in section B.3.
>
> > It is also good to mention whether it can be applied to systems with friction.
>
> Our framework can be extended to systems with friction in a relatively straight forward way, by adding a dissipation term to the derivative function generated by the Hamiltonian, in a similar way to Tanaka et al. (2022).  We did implement this, and carried out some initial experiments as part of the project, but chose to leave it out of the paper due to time and space constraints and for ease of exposition. Note that in the case of friction, the energy conserving shooting method is not applicable, and one must use shooting with no energy constraint term.
>
> > I am having trouble understanding the relationship between the variables s,y,x. Please clarify.
>
> A further explanation of each variable is given below. We apologise for the lack of clarity here.
>
> * The variables $\mathbf{s}$ represent the shooting states for the model, that is to say they represent the initial conditions for each short shooting segment, that taken together represent the whole trajectory that is to be learned. The shooting states must be inferred and we use VI to compute their approximate posterior. As discussed in the paper, it is important that the shooting states are matched to adjacent sections both in terms of Euclidean distance and distance in energy space, for the trajectory to be consistent with that generated from a Hamiltonian system.
> * The variable $\mathbf{x}(t)$ represents the latent noise-free trajectory of the system, which we aim to infer, at time $t$. $\mathbf{x}(t)$ is generated by integrating the derivative function forward over each segment, with the derivative function being generated from the hamiltonian.
> * The variables $\mathbf{y}$ are the data we observe, which we consider to be realisations of the systems trajectory $\mathbf{x}$ corrupted with additive Gaussian distributed noise.
>
> We are happy to put this in the update manuscript if that useful. Please let use know if we should explain further.
>
> > Please explain in detail how to sample the initial conditions when generating data in an experiment. Do you limit the energy range, etc.?
>
> The sampling procedure for the initial conditions are as follows for each system:
> - **Fixed pendulum** We sample initial conditions by drawing $p, q \sim U (−1, 1)$, and discard those with energy $E > mgr$ to avoid the pendulum swinging over its pivot.
> - **Spring pendulum** We sample initial conditions as $q_1,q_2,p_1,p_2 ∼ U(−0.25,0.25)$ and do not limit the energy range.
> - **Henon-Heiles** We sample initial conditions by first sampling $q_1,q_2,p_1,p_2\sim U(-1, 1)$ and then discarding initial conditions with $E>\frac{1}{6\mu^2}$ to ensure trajectories stay in the region of phase space with connected level sets, that is to say the trajectories stay localised after long times, and do not tend to infinity, see Offen and Ober-Blöbaum (2022) for more details.
>
> These details were also listed in the appendix (B.1),  but we can add expanded discussion if necessary.
>
> ---
>
> If we have been unclear on any of the points above, or you would like further clarification on anything, please let us know.

---

### Review · Reviewer_GDmG · 2022-12-20

**Summary Of Contributions:**

The manuscript presents a Gaussian process model for Hamiltonian systems with decoupled parameterization and introduces an energy-conserving shooting method that allows inference from dynamics trajectories. The paper then compares the performances of the approach in learning Hamiltonian systems with existing methods in various data settings.

**Audience:**

Yes

**Claims And Evidence:**

No

**Requested Changes:**

- [Optional] Inducing variables and inducing points (Section 3.2) are mentioned as an established concept. I suggest adding some relevant background to make it accessible to a general audience with less exposure to the Gaussian process.
- [Typo] Page 2: In this work
- Figure 1: There shouldn’t be an abbreviation in the legend. What are inducing energies?

**Strengths And Weaknesses:**

### Strengths
- The paper addresses a problem that might be of broad interest in machine learning and applied mathematics.
- In general, the paper is well-written with appealing visualizations.
- The technical developments of the work are rigorous and intuitive. Background information and relevant literature are adequately covered.


### Weaknesses

1. In my opinion:
    - The paper overstates the significance of its technical contributions to inference methodology.
    - The current presentation of the paper doesn’t highlight well enough its connection to the closest reference, Hegde et al. (2022). This might give the readers a false sense of novelty.

    As pointed out by the authors (but very late into the manuscript), in terms of methodology, the work depends heavily on Hegde et al. (2022), which introduces variational multiple shooting for general ODEs using Gaussian processes. As far as I can tell, the technical constructions of the two papers are the same, and the only major algorithmic adjustment of the paper is to hardcode the conservation of energy into the distribution (Equation 27-28) to specifically target Hamiltonian systems.

    While this idea is intuitive and reasonable, given the close relation of the two frameworks, I suggest that the authors:

    - remove their claims of significance to the inference methodology
    - restructure the framing of the work to explicitly acknowledge Hegde et al. (2022) (early, in the Methodological sections, not Related works) as the foundation of the framework. I also suggest that the authors highlight in each part of the framework the deviations of the methods from Hegde et al. (2022), and possibly explain why such changes are made/necessary.


2.  Experiments: I find the experimental evaluations of the work unconvincing:
    - Since the paper follows the framework of Hegde et al. (2022) with an additional hardcoding of the law of conservation of energy, it is expected that the method will outperform Hegde et al. (2022) for Hamiltonian systems. The main question for the evaluations is whether the method is comparable with pre-existing Hamiltonian Gaussian Process baselines.
    - It is thus disappointing that such comparisons are not available:
        - “A public code exists for the SSGP, however at the time of writing it does not support experiments with a single trajectory, or systems with D > 1.” Why don’t we test and compare the performance on multiple trajectories, or for D=1?
        - “Code for the SPGP is not publicly available”: It seems that the paper SPGP stated that code will be published upon request. Have the authors reached out to SPGP’s authors for access?

        I’m aware that the paper attempt to use HGP-Batched to emulate SSGP/SPGP, but comparisons with a proxy is less convincing and should only be done if there are no alternatives.

---

> ### Author Response · Authors · 2022-12-21
> **Acknowledging review**
>
> We thank you for your review, and especially for the fast turnaround at this time of year. We would like to let you know that we have seen the review, but due to previous commitments over the holiday period, we will not be able to respond until approximately the 13th of January. This should still be well within the prescribed timeframe (4 weeks after all 3 reviews posted).  Thanks.

---

> > ### Comment · Reviewer_GDmG · 2023-01-30
> > **Comments addressed**
> >
> > I want to thank the authors for a very detailed response, and I want to show my admirations for their efforts to address my comments and to revise the manuscript. All of my comments from the original review are now addressed. I want to congratulate the authors for a nice manuscript, and it has been my pleasure working with them for this revision.

---

> ### Author Response · Authors · 2023-01-13
> **Response to Reviewer GDmG: Weakness 1**
>
>
> We thank the reviewer for their time and the fair and thorough evaluation of our work. We address the two weaknesses raised below.
>
> #### Weakness 1: Claimed contribution/significance
>
> We appreciate the review pointing out the problems with the framing relative to the GPODE model (Hegde et al 2022) and, on reflection, agree that our framing of the work made it unclear exactly which parts of the model and inference methodology were novel. We never intended to mislead the reader into thinking we made contributions that we had not, and apologise if it came across that way.
>
> Following your suggestions we’ve added extended descriptions of how our method relates to the GPODE throughout section 3, as well as adjusting our claimed contribution in the introduction. We have updated the manuscript, and new content is shown in blue. Mainly, we clearly acknowledge that our work builds on the multiple shooting variational inference of Hegde et al, and we point out that we include a shooting energy constraint, and place inducing points on the energy instead of the differential. We also reframe the inference contribution #2 (in introduction) similarly. We now believe the claimed contributions and discussion of methodology faithfully represent the relationship between the models, but if you still feel this not the case, please let us know and we will try to make a larger adjustment to the framing.

---

> ### Author Response · Authors · 2023-01-13
> **Response to Reviewer GDmG: Weakness 2 (Part 1, discussion of competing methods)**
>
> ### Weakness 2: Comparison to SPGP/SSGP baselines
>
> We understand the reviewers concerns about comparison with competing GP methods, the SSGP and SPGP.  We first offer some further explanation for why we did not offer a comparison with these methods, and then discuss some results obtained from the SSGP.
>
> **SPGP**: The SPGP integrates GPs into symplectic Hamiltonian integrators for numerical robustness. The method is non-trivial to implement, and the implementation is not publicly available. We reached out to the authors, and they responded that the implementation is not available even by request.
>
> **SSGP**: The SSGP is concurrent to our work, and also designs a GP model for Hamiltonian systems. The main difference is that they do not use inducing points, but opts for random Fourier inference, and do not use shooting.
>
> The SSGP focuses on task 2 (initial condition extrapolation) with large number (10..50) of short trajectories. They do not consider task 1 at all. Our focus is mainly on efficient inference for single (or few) long trajectories using the shooting formulation, although we do evaluate our model on task 2 as well. Additionally we wished to go beyond simple single dimensional systems, and evaluate our model on higher dimensional systems with more complex dynamics, which is not possible with the implementation of the SSGP provided by the authors, which unfortunately only works for systems with $D=1$. Finally, the SSGP implementation provided by the authors was very much still a work in progress, with little to no documentation or comments, with hardcodings, and no interface for providing ones own data, making it difficult to use out of the box for experimental comparisons. These reasons together caused us to decide to leave the comparison out of the paper. We now see that this was an oversight, and we should have included a comparison on the experiments for which this is possible, the results of this comparison are below.
>
> We do believe that the models we included for comparison were relevant, as it is important to see how performance compares to models that do not make Hamiltonian assumptions (GPODE, NODE), and the typical NN model used for problems of this type (HNN).

---

> ### Author Response · Authors · 2023-01-13
> **Response to Reviewer GDmG: Weakness 2 (Part 2, SSGP results)**
>
> ##### SSGP results
>
> After spending time making some small adjustments to the author provided code for SSGP, including data ingestion and fixing a minor bug that was causing the code to fail for problems with a single trajectory, we were able to generate results for task 1 and 2 for the FP system. These results are shown below, along with the other models discussed in the paper for comparison.
>
> | Method | State RMSE | State MNLL | Energy RMSE |
> | ------ | ---------- | ---------- | ----------- |
> | NODE   | **0.12 (0.12)** | -                | **0.30 (0.28)** |
> | HNN    | 0.20 (0.16)     | -                | **0.34 (0.33)** |
> | GPODE  | 0.23 (0.26)     | -0.03 (1.21)     | **0.38 (0.34)** |
> | HGP    | 0.18 (0.19)     | **-0.21 (0.67)** | **0.25 (0.42)** |
> | SSGP   | 1.49 (0.39)     | 1.50 (0.20)      | **0.23 (0.32)** |
>
> **Task 1**: As expected, the SSGP model performs poorly on the trajectory forecasting task, which mirrors our findings in our comparison with the HGP-Batched model, discussed in the paper. In terms of both RMSE and MNLL performance the HGP produces considerable better results. On this task the best model is the NODE, which gives slightly better results than the HGP.
>
> | Method | State RMSE | State MNLL | Energy RMSE |
> | ------ | ---------- | ---------- | ----------- |
> | NODE   | **0.48 (0.15)** | -               | 0.26 (0.06)     |
> | HNN    | 0.91 (0.47)     | -               | 0.25 (0.15)     |
> | GPODE  | 2.41 (2.28)     | 7.11 (8.22)     | 6.17 (10.30)     |
> | HGP    | 1.41 (0.46)     | 4.68 (2.95)     | 0.55 (0.60)     |
> | HGP-Batched |**0.54 (0.31)** |**0.66 (0.51)** |**0.10 (0.03)** |
> | SSGP   | 0.75 (0.50)     | 0.89 (0.37) | 0.13 (0.03)|
>
> **Task 2**: The SSGP model performs well for task 2 in comparison to the shooting HGP, which is perhaps not surprising given that this is the task that SSGP focused on ($D=1$, multiple trajectories). The SSGP performs considerably worse than NODE in terms of state RMSE, although in terms of energy RMSE it provides a good result. In order to determine the cause of the performance difference between SSGP and HGP with shooting, we also ran the HGP model with batching. We found the HGP with batching provided better performance than the SSGP and HGP with shooting on this task. This indicates that the performance difference between the shooting HGP and the SSGP on task 2 can be attributed to the batching vs shooting, and implies that the other differences between the models (inference via inducing points vs RFFs, inferring the initial conditions, Hamiltonian aware initialisation) have a positive effect on performance.
>
> Overall these results show that for systems with $D=1$, the HGP with shooting provides significantly better trajectory forecasting performance, whereas the batching-based inference provides better extrapolation to new initial conditions, from a given set of initial conditions. We discuss some points raised by these results below:
>
> - These results are for systems with $D=1$, it is not clear that the improved performance of the SSGP for task 2 would extend to more complex systems. In all experiments the relative performance of the HGP compared to baselines is improved for the more complex systems, this may well also be the case for the HGP performance relative to the SSGP.
> - The results for task 2 show that shooting based inference is likely non optimal for the problem of trajectory extrapolation with multiple trajectories. Fitting multiple trajectories jointly is a more difficult optimisation problem, since the inducing points must be effectively distributed over a larger area of phase space. Adding the shooting objectives for this problem narrows down the space of feasible solutions and makes the optimisation problem more difficult, making it harder to find a good solution. We plan to investigate this effect further as part of future work.
> - We note the runtimes of both models:  to generate the set of results for the HGP on the FP system takes around an hour, with the SSGP taking around 24 hours to generate the same results.
>
> We have added a these results along with the discussion to section C.3 of the manuscript.
>
> ---
>
> We hope our response has provided sufficient fulfillment of the acceptance criteria, especially regarding the “Claims and Evidence” criteria. Please let us know if there are further concerns.

---

### Review · Reviewer_xxKa · 2023-01-25

**Summary Of Contributions:**

Sincerest apologies to the editors and the authors for handing in this review late. I hope that the review is still helpful. If so, I would like to request that the authors be granted some extra time to respond to the review. I would also like to add that I am reviewing the revised version of the manuscript.

This work consider the task of learning Hamiltonian systems from a few long-horizon trajectories via Gaussian processes. It infers inducing points from observational data by variational inference; ELBO is maximized by using a shooting method to avoid blow up of gradients on long trajectories.


**Audience:**

Yes

**Claims And Evidence:**

Yes

**Requested Changes:**

Direct comparisons to the experiments in Hegde et al are difficult since the experiments therein involve non-conservative dynamics, i.e. there is no potential function. However, in the present context, it would helpful to know which of the two primary modifications (direct prior over Hamiltonian, or energy constraints in shooting) is responsible in greater part for the empirical gains overs GPODE.

Disentangling contributions -- neural net vs GP, energy-conserving, imposing Hamiltonian -- in empirical terms would certainly add value to the present work.

**Strengths And Weaknesses:**

The overall scheme is similar to the proposal of Hegde et al, where learning potentially non-conservative vector fields is considered. The two primary differences here are that (1) the authors propose considering inducing points on the Hamiltonian rather than the force field (recall, there's no Hamiltonian in the other work), (2) add energy-conservation terms at the end points of the segments in the shooting method for inference (again, only makes sense to conserve energy for conservative fields). Although the rest of the recipe is similar, the work demonstrates impressive gains in comparison to GPODE, and perhaps less convincingly with respect to other comparators.

---

> ### Author Response · Authors · 2023-01-27
> **Response to Reviewer xxKa**
>
> We thank the reviewer for the review. No apology is necessary for the lateness, we understand this is a busy time of year and appreciate the feedback that has been offered. We address the reviewer's points below.
>
> > It would helpful to know which of the two primary modifications (direct prior over Hamiltonian, or energy constraints in shooting) is responsible in greater part for the empirical gains overs GPODE
>
> We can compare the effect of the Hamiltonian prior vs the energy constrained shooting by looking at the performance gains over the GPODE on the experiments in task 1, and observing the relationship between the performance of the HGP with and without energy conserving shooting (HGP-S vs HGP-ES) in Figure 7a. The inclusion of the Hamiltonian structure provides a larger improvement in performance, with the addition of energy conservation to the shooting providing a more modest gain. We should note that it is not possible to have energy conserving shooting without having a Hamiltonian prior, since it is this prior that provides the constraint. We can see from the results in 7a that given a Hamiltonian prior, the inclusion of energy conserving shooting does improve performance further, but to a lesser extent than the Hamiltonian prior.  If necessary, we can provide additional experiments to further illustrate this point, please let us know if so, however we believe the results we have already demonstrate this. We can also add further discussion to the manuscript if necessary.
>
> > Disentangling contributions -- neural net vs GP, energy-conserving, imposing Hamiltonian -- in empirical terms would certainly add value to the present work.
>
> In the experiments section we have offered a comparison with both NN and GP models that both include the Hamiltonian prior and do not, which we believe allow us to demonstrate empirically the contributions of each. As discussed in section 7, we interestingly found that the HNN, with the additional bias over the NODE actually performed worse. This is likely due to the additional restrictions enforced by the HNN for a problem where training NNs models is already difficult and unstable. Throughout, the HGP outperformed the GPODE, indicating the contribution of the Hamiltonian prior for the GP models. Results on task 2 indicate the stationary GP prior may not be the most suitable for the problem of extrapolation to new trajectories, and investigation into nonstationary priors may be necessary. On the other hand, the GP prior allows smooth Hamiltonians and differentials to be encoded, while in neural networks this is not enforced without special treatments, this can be seen most clearly in the animations provided in the supplementary material, where the NN models generate trajectories, which are clearly unphysical. An optimal prior would retain the smoothness, whilst also accounting for nonstationarity.
>
> ---
> If you have any additional requests, please let us know.

---

### Author Response · Authors · 2023-01-13
**General response to reviewers**

We thank the reviewers for their time. Although we have not yet received all three reviews, we have decided to respond now, as it has already been a significant amount of time since the initial reviews were posted. Below we answer each review separately. We have also updated the pdf manuscript with corresponding new changes, clearly marked with blue color.  It may be the case that when the final review is posted further updates are required, if so, we will post again here.

---

### Decision · Action_Editors · 2023-02-24

**Recommendation:** Accept as is

**Comment:**

The authors have already done a good job in the initial submission and in the responses tot he referees. They are welcome to works those responses into the final version of the paper.

**Audience:**

The paper presents a nice development within the topic of using Gaussian processes for learning Hamiltonian mechanics. This is relevant for many applications within machine learning and natural sciences.

**Claims And Evidence:**

The reviewers all agree that the paper presents a substantial contribution, the authors have made an effect to address concerns and that the paper is in very good shape. Acceptance with no revisions are recommended other than those the authors have already made or minor voluntary revisions that the authors plan to make.